# Angel or Demon: Investigating the Plasticity Interventions' Impact on Backdoor Threats in Deep Reinforcement Learning

**Oubo Ma** [1]  **Ruixiao Lin** [1]  **Yang Dai** [2]  **Jiahao Chen** [1]  **Chunyi Zhou** [1]  **Linkang Du** [3]  **Shouling Ji** [* 1]

## Abstract

Extensive research has highlighted the severe threats posed by backdoor attacks to deep reinforcement learning (DRL). However, prior studies primarily focus on vanilla scenarios, while plasticity interventions have emerged as indispensable built-in components of modern DRL agents. Despite their effectiveness in mitigating plasticity loss, the impact of these interventions on DRL backdoor vulnerabilities remains underexplored, and this lack of systematic investigation poses risks in practical DRL deployments. To bridge this gap, we empirically study 14,664 cases integrating representative interventions and attack scenarios. We find that only one intervention (i.e., *SAM*) exacerbates backdoor threats, while other interventions mitigate them. Pathological analysis identifies that the exacerbation is attributed to backdoor gradient amplification, while the mitigation stems from activation pathway disruption and representation space compression. From these findings, we derive two novel insights: (1) a conceptual framework *SCC* for robust backdoor injection that deconstructs the mechanistic interplay between interventions and backdoors in DRL, and (2) abnormal loss landscape sharpness as a key indicator for DRL backdoor detection.

## 1. Introduction

Deep Reinforcement Learning (DRL) has achieved widespread adoption in safety-critical applications, including robotic control (Wang et al., 2024), drone navigation (Kaufmann et al., 2023), and autonomous driving (Tang et al., 2025). However, DRL agents are vulnerable to severe security threats from backdoor attacks (Rathbun et al., 2024;

[1]Zhejiang University [2]National University of Defense Technology [3]Xi'an Jiaotong University. Correspondence to: Shouling Ji <sji@zju.edu.cn>.

*Proceedings of the 43rd International Conference on Machine Learning*, Seoul, South Korea. PMLR 306, 2026. Copyright 2026 by the author(s).

Liu et al., 2025), which embed malicious trigger-action mappings that result in catastrophic failures. For instance, adversaries can exploit backdoors to force autonomous vehicles into abrupt maneuvers, resulting in traffic congestion or collisions (Chen et al., 2025).

Existing DRL backdoor research primarily emphasizes enhancing attack techniques (e.g., transition tampering (Kiourti et al., 2020; Dai et al., 2026) and backdoor reward exploration (Ma et al., 2025; Rathbun et al., 2025)), yet victim agents are trained and evaluated on vanilla DRL pipelines. Concurrently, a cutting-edge research field addresses plasticity loss in DRL, which arises from its inherent properties of non-stationary input streams and shifting optimization objectives (Dohare et al., 2024; Lyle et al., 2024; Ma et al., 2024a). These studies suggest that DRL agents typically incorporate *plasticity interventions* to sustain their continuous learning capability, with interventions integrated as integral components within DRL implementation, such as *Shrink & Perturb* (Ash & Adams, 2020), *Weight Clipping* (Elsayed et al., 2024), *Spectral Normalization* (Gogianu et al., 2021), *Weight Decay* (Dohare et al., 2024), *Layer Normalization* (Lyle et al., 2023), *ReDo* (Sokar et al., 2023), and *SAM* (Lee et al., 2023).

Despite significant research efforts in these respective fields, the impact of interventions on DRL backdoor attacks remains largely underexplored, particularly regarding whether they alter the internal properties of backdoors, thereby posing realistic risks in practical DRL deployments. To bridge this gap, we conduct a comprehensive investigation following the evaluation framework outlined in Figure 1, aiming to answer the following Research Questions (RQs):

***RQ1: How do plasticity interventions impact backdoor attacks in DRL?*** We conduct a comprehensive empirical evaluation encompassing 9,024 cases, integrating seven prominent plasticity interventions in representative attack scenarios. Each attack scenario comprises a DRL task, an attack phase (learning-from-scratch vs. post-training), an attack method, and a backdoor task. The results reveal that interventions in the post-training phase have a more pronounced impact on DRL backdoors. Notably, only *SAM* exacerbates backdoor risks, whereas the other interventions exhibit varying degrees of mitigation. For instance, in robotic control

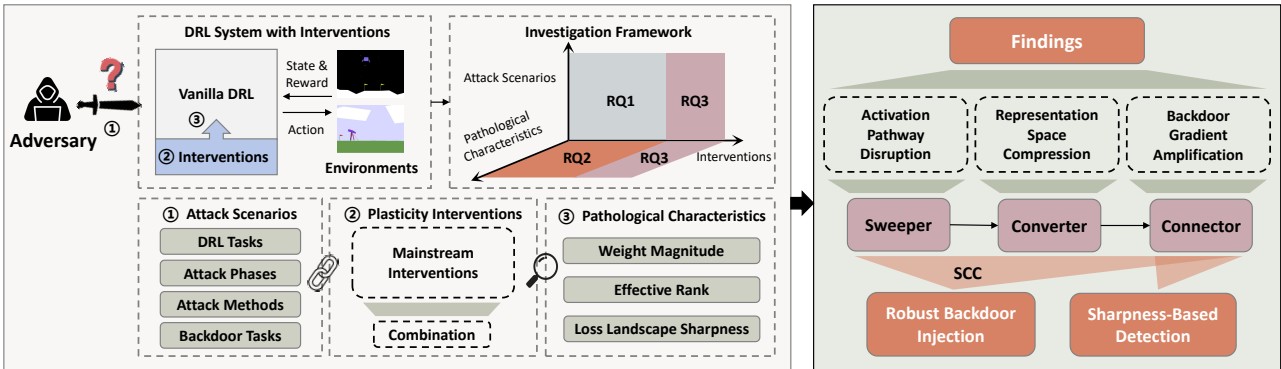

*Figure 1.* Outline of our investigation.

tasks, the application of *SAM* raises the attack success rate (ASR) from 0.178 to 0.326 and simultaneously enhances benign task performance (BTP) from 0.745 to 0.814.

***RQ2: What internal properties of DRL backdoors do interventions alter to drive these impacts?*** Extending the pathological analysis from the plasticity loss domain, we attribute the observed effects in RQ1 to three distinct mechanisms: **(M1)** Activation pathway disruption[1] induces competition between backdoor and benign tasks (e.g., *Shrink & Perturb*, *Weight Clipping*, and *ReDo*). **(M2)** Representation space compression drives backdoor and benign gradients from orthogonality toward alignment, creating denser backdoor pathways and exacerbating non-stationarity (e.g., *Spectral Normalization*, *Weight Decay*, *Layer Normalization*). **(M3)** Backdoor gradient amplification, by capturing sharp losses, enables the backdoor pathway to rapidly converge to a flat-minimum region that is robust to parameter perturbations (e.g., *SAM*).

***RQ3: What novel insights emerge from our findings for advancing DRL backdoor research?*** Through an additional investigation of 5,640 cases, we discover that combining multiple interventions tends to amplify backdoor threats rather than mitigate them. For instance, in robotic control tasks, ASR increases from 0.178 to 0.418, while BTP increases from 0.745 to 0.915. Leveraging this insight alongside the internal properties identified in RQ2, we propose a conceptual framework for robust backdoor injection in post-training scenarios, termed *Sweeper-Converter-Connector* (*SCC*): *Sweeper* exploits **M1**, which denotes clipping or resetting weights to vacate neural pathways, thereby creating space for backdoor injection. *Converter* leverages **M2**, which conceptualizes aligning backdoor and benign gradients, transforming backdoors into multi-pathway structures. *Connector* applies **M3**, which guides optimization toward flat minima, stabilizing the co-construction of backdoor and benign representations across multiple pathways. *SCC* also

enables a pre-deployment diagnosis, especially instrumental for the emerging deployment of integrating multiple interventions in the backdoor-sensitive scenarios. Meanwhile, we reveal that abnormal loss landscape sharpness consistently manifests as a distinctive signature of DRL backdoors across both learning-from-scratch and post-training scenarios, offering a promising avenue for backdoor detection.

In summary, this study makes the following contributions:

- We conduct the first comprehensive investigation (covering 14,664 cases) into how seven mainstream interventions and five combination strategies impact DRL backdoor attacks across diverse attack scenarios.

- We identify three mechanisms by which interventions alter the internal properties of DRL backdoors: activation pathway disruption, representation space compression, and backdoor gradient amplification.

- We derive two novel insights for advancing DRL backdoor research regarding backdoor internal properties (i.e., *SCC* for robust injection) and external manifestations (i.e., sharpness-based detection). Additionally, we release the source code to facilitate reproducibility[2].

## 2. Background

This section summarizes the background relevant to this study. Additional details are provided in Appendix A.

**DRL Backdoor Attacks.** Existing backdoor attacks follow a two-step pipeline (i.e., backdoor injection and backdoor activation): the backdoor is first injected during the agent's training phase, and is later activated during deployment to enable unauthorized manipulation of the agent's behavior. The primary technique for backdoor injection is transition tampering (Kiourti et al., 2020; Dai et al., 2026), where the adversary modifies the transitions (i.e., triplets consisting

---

[1]An activation pathway is conceptualized as a functional subnetwork formed by parameter connections.

[2]https://github.com/maoubo/Plasticity

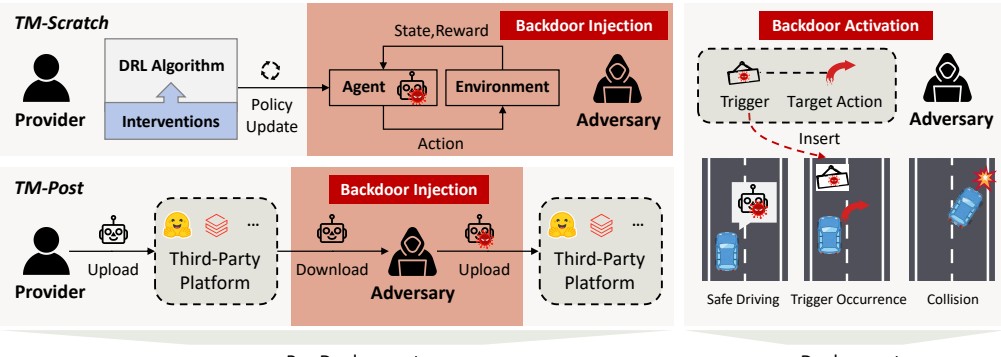

*Figure 2.* Threat models.

of state, action, and reward) stored by the agent to bind the trigger with the target action through backdoor rewards. Additional injection techniques include environment perturbation (Yang et al., 2019; Liu et al., 2025) and policy combination (Wang et al., 2021; Gong et al., 2024). The adversary can further escalate the backdoor threat by targeting both triggers and rewards. Techniques such as trigger optimization (Cui et al., 2024; Li et al., 2025) and reward modification (Rathbun et al., 2024; 2025) help alleviate update conflicts, whereas backdoor reward exploration (Ma et al., 2025) improves the cross-environment applicability of DRL backdoor attacks.

**Plasticity Interventions.** Interventions are designed to preserve stable input representations and training dynamics, which is crucial for the practical utility of DRL agents. The design of mainstream plasticity interventions is primarily motivated by four intuitions (Klein et al., 2024): weight perturbation, weight regularization, activation control, and training control. Weight perturbation (Ash & Adams, 2020; Elsayed et al., 2024; Hernandez-Garcia et al., 2025) involves directly clipping or perturbing the parameter weights of the policy to mitigate the adverse effects of outlier weights on the training dynamics. Weight regularization (Gogianu et al., 2021; Lyle et al., 2022; Dohare et al., 2024) applies soft constraints on weight updating to encourage the exploration of the policy in parameter space. Activation control (Nikishin et al., 2022; Lyle et al., 2023; Sokar et al., 2023; Abbas et al., 2023) involves regulating intermediate activations (e.g., normalizing activations and modifying activation functions) to reduce the sensitivity of representations to non-stationary inputs. Training control (Lee et al., 2023; Nikishin et al., 2023; Lee et al., 2024; Ma et al., 2024a) modifies the optimization process or objective to steer policy updates in a more stable direction, reducing the risk of aggressive updates that could lead to suboptimal solutions due to environmental dynamics. In addition, recent studies suggest that combining different interventions has the potential to further reduce the plasticity loss of the policy (Lyle et al., 2024; 2025).

## 3. Problem Formulation

**Threat Model.** The attack scenario involves a provider and an adversary (see Figure 2). The provider trains the DRL agent from scratch for the benign task and determines whether interventions should be applied to improve the agent's continual learning capability. Then, the provider deploys the well-trained agent or uploads it to a third-party platform. Based on the stage at which the adversary initiates the backdoor injection, we consider two widely discussed threat models (Kiourti et al., 2020; Cui et al., 2024; Rathbun et al., 2024; Ma et al., 2025; Dai et al., 2026):

*TM-Scratch. The adversary injects the backdoor while the provider is training the agent.*

*TM-Post. The adversary downloads the released agent, injects a backdoor via post-training, and then republishes the backdoored agent to the third-party platform.*

During the deployment phase, the backdoored agent performs sequential decision-making normally on the benign task. However, when the adversary inserts a trigger into the environment, the backdoor is activated, forcing the agent to output the target action. Appendix B provides supplementary clarification of the threat model and a detailed description of the backdoor injection process.

**Formulation.** The benign task is modeled as a Markov Decision Process, denoted by $\mathcal{M} = (\mathcal{S}, \mathcal{A}, \mathcal{R}, \mathcal{P}, \gamma)$, where $\mathcal{S}$ is the state space, $\mathcal{A}$ is the action space, and $\mathcal{R}$ is the reward function. The state transition function $\mathcal{P} : \mathcal{S} \times \mathcal{A} \to \Delta(\mathcal{S})$ defines the probability of reaching state $s' \in \mathcal{S}$ after taking action $a \in \mathcal{A}$ in state $s \in \mathcal{S}$. The discount factor $\gamma \in [0, 1)$ balances immediate and future rewards. The policy of the agent $\pi_\theta : \mathcal{S} \to \Delta(\mathcal{A})$ maps each state to a probability distribution over actions. The DRL algorithm aims to find the optimal parameters for the policy $\pi_\theta$ by maximizing the expected cumulative reward, i.e., $\theta^* = \arg\max_\theta \mathbb{E}_{\pi_\theta}[\sum_{t=0}^{T} \gamma^t r_t]$, where $T$ is the time horizon.

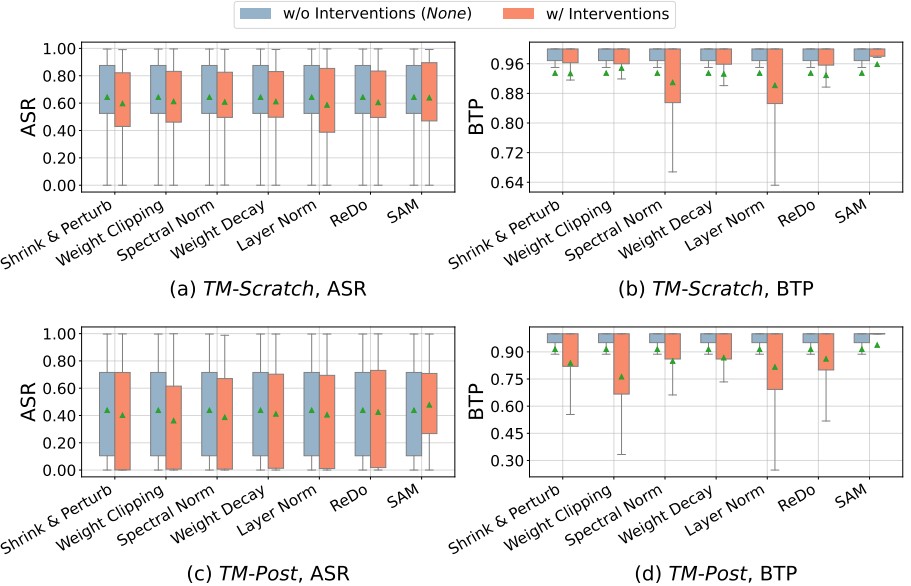

Figure 3. Impact of interventions in *TM-Scratch* and *TM-Post*.

The backdoor task is modeled as a tuple $\mathcal{M}^\dagger = (\mathcal{T}, \mathcal{S}^\dagger, \mathcal{A}^\dagger, \mathcal{F}_s, \mathcal{F}_a, \mathcal{R}^\dagger)$, where $\mathcal{T}$ is the trigger space, $\mathcal{S}^\dagger \subseteq \mathcal{S}$ is the backdoor state space, and $\mathcal{A}^\dagger \subseteq \mathcal{A}$ is the target action space. $\mathcal{F}_s : \mathcal{S} \times \mathcal{T} \to \mathcal{S}^\dagger$ is a trigger-state mapping function that defines how a trigger alters a benign state. $\mathcal{F}_a : \mathcal{T} \to \mathcal{A}^\dagger$ is a trigger-action mapping function specified by the adversary, which determines the target action $a^\dagger \in \mathcal{A}^\dagger$ corresponding to each trigger $\delta \in \mathcal{T}$. $\mathcal{R}^\dagger$ is a backdoor reward function crafted to reinforce the mapping between triggers and target actions.

**Adversary's Objective:** the objective of the adversary is to induce the agent to output an action that is as close as possible to the predefined target action when the backdoor is activated, i.e., $\min_{\theta^\dagger} \mathbb{E}_{s \sim \mathcal{S}, \delta \sim \mathcal{T}} \left[ ||\pi_{\theta^\dagger}(\mathcal{F}_s(s, \delta)) - a^\dagger|| \right]$, where $\pi_{\theta^\dagger}$ is the backdoored policy. Meanwhile, the adversary seeks to avoid degrading the agent's performance on the benign task.

## 4. Study Design

**Attack Scenarios.** We construct diverse attack scenarios to underpin a comprehensive investigation. • For DRL tasks (i.e., benign tasks), we adopt four classic control tasks (CartPole, Acrobot, MountainCar, and Pendulum) and two physics control tasks (Lunar Lander and BipedalWalker) from OpenAI Gym (Brockman et al., 2016), as well as three robotic tasks (Hopper, Reacher, and Half Cheetah) from Facebook AI's PyBullet (Coumans & Bai, 2021). These tasks span discrete and continuous action spaces, sparse and dense reward signals, and both cold-start and non-cold-start conditions. • We consider performing backdoor injec-

tion during both the learning-from-scratch and post-training stages (i.e., *TM-Scratch* and *TM-Post*). • The backdoor attacks are carried out using four representative methods, all of which are compatible with both *TM-Scratch* and *TM-Post*: TrojDRL (Kiourti et al., 2020), BadRL (Cui et al., 2024), SleeperNets (Rathbun et al., 2024), and UNIDOOR (Ma et al., 2025). • We construct 47 backdoor tasks with reference to Ma et al. (2025), including both single-backdoor and multi-backdoor injection settings.

**Intervention Setup.** We consider eight intervention settings. • *None*: no intervention is applied, serving as a baseline for comparison; • *Shrink & Perturb*: upon each network update, weights are scaled by a small scalar and perturbed by adding weights from a randomly initialized network; • *Weight Clipping*: constraining the weights to lie within a predefined range; • *Spectral Normalization*: applied after the initial linear layer of the network; • *Weight Decay*: setting the $\ell_2$ penalty coefficient to $10^{-5}$; • *Layer Normalization*: applied after every linear layer; • *ReDo*: periodically resetting the neurons with the highest dormancy level in each layer at fixed intervals; • *SAM*: applying Sharpness-Aware Minimization with Adam as the base optimizer, setting the sharpness penalty to 0.01. We formalize these intervention settings as a set $P = \{p_1, p_2, \ldots, p_8\}$, where each $p_i$ corresponds to the $i$-th intervention setting introduced above (e.g., $p_8$ denotes *SAM*). We also consider two existing combinations, *Swiss Cheese* (Lyle et al., 2024) and *Plastic* (Lee et al., 2023), alongside three new combinations based on the aforementioned interventions: *Lac*, *SLac*, and *SSW*.

**Pathological Characteristics.** Prior studies indicate that interventions modulate a DRL agent's continual learning

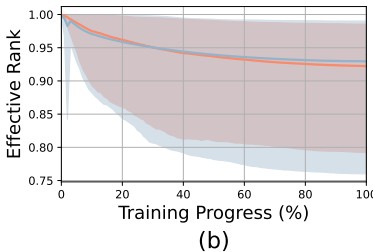 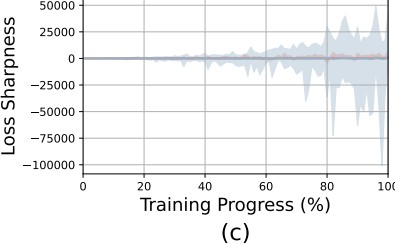

*Figure 4.* Comparison of conventional training and backdoor attacks on the three pathological characteristics. Solid lines represent mean values, and shaded areas denote the range from minimum to maximum values. — Conventional Training, — Backdoor Attacks.

ability via three pathological characteristics: weight magnitude, effective rank, and loss landscape sharpness (Lyle et al., 2023; Sokar et al., 2023; Dohare et al., 2024; Klein et al., 2024). In line with these studies, we analyze how interventions affect DRL backdoor attacks through these pathologies. We formalize these pathologies as a set $C = \{c_1, c_2, c_3\}$, where each $c_i$ corresponds to the $i$-th pathology introduced above (e.g., $c_1$ denotes weight magnitude).

Appendix C provides additional details on definitions (e.g., combinations, pathological characteristics and evaluation metrics) and implementations (e.g., DRL algorithms, backdoor attacks, and benign & backdoor task designs).

## 5. Empirical Study and Analysis

This section systematically explores the three research questions and derives five key findings.

### 5.1. RQ1: Impact of Interventions

The reported results encompass 9,024 experimental cases, derived from 2 threat models × 8 intervention settings × 47 backdoor tasks × 4 backdoor attacks × 3 random seeds.

Figure 3 (a) and (b) show that in *TM-Scratch*, interventions exert a modest impact on DRL backdoor attacks. Specifically, Figure 3(a) reveals that ASR exhibits minor fluctuations, with *Layer Normalization* yielding the most pronounced effect of -8.84%. Figure 3(b) indicates that *Spectral Normalization* and *Layer Normalization* lead to more pronounced fluctuations in BTP. For instance, *Layer Normalization* reduces BTP by over 30% in the Acrobot and Mountain-Car tasks, both of which are characterized by sparse rewards. Notably, *SAM* is the only intervention that slightly improves BTP while maintaining ASR virtually unchanged. Detailed numerical results are provided in Table 10 in the Appendix. Furthermore, Appendix D demonstrates that these interventions have a negligible impact on BTP under conventional training. Therefore, the BTP fluctuations observed in Figure 3(b) are attributable to the effects of the interventions on DRL backdoor attacks.

Figure 3 (c) and (d) show that in *TM-Post*, interventions exert a more pronounced impact on DRL backdoor attacks. This is because interventions affect a well-trained DRL agent (suffering from plasticity loss) more substantially than a randomly initialized agent. Figure 3(c) indicates that most interventions reduce ASR; specifically, *Weight Clipping* decreases it by an average of 17.46% and *Spectral Normalization* by 11.78%. Figure 3(d) indicates that most interventions significantly reduce BTP, with *Weight Clipping* decreasing it by an average of 20.19% and *Layer Normalization* by 11.93%. Remarkably, we observe that in *TM-Post*, *SAM* produces a pronounced enhancement of backdoor attacks. For instance, in robotic tasks, it increases ASR from 0.178 to 0.326, representing an 83.15% relative improvement, while in physics control tasks, the gain reaches 26.07%. Detailed numerical results are provided in Table 11 in the Appendix. Moreover, the ablation study in Appendix E demonstrates that despite fluctuations caused by different intervention hyperparameters, these effects exhibit a consistent trend.

> **(Finding 1) Heterogeneity Impact**: Interventions exert a more substantial impact on DRL backdoor attacks in *TM-Post* than in *TM-Scratch*. Notably, *SAM* significantly exacerbates the backdoor threat, while other interventions exhibit varying degrees of mitigation.

### 5.2. RQ2: Intrinsic Mechanisms

In this subsection, we first examine the impact of backdoor attacks on the agent with respect to the three pathological characteristics, establishing a baseline for subsequent comparison with interventions. Figure 4 demonstrates that backdoor attacks substantially amplify the non-stationarity of DRL training, as evidenced by larger performance oscillations. Specifically, the performance ranges (i.e., the absolute differences between the maximum and minimum values) of weight magnitude and effective rank increase by 98.63% and 19.16%, respectively. The most pronounced effect is observed in loss landscape sharpness, the range of which increases by **635.22%**.

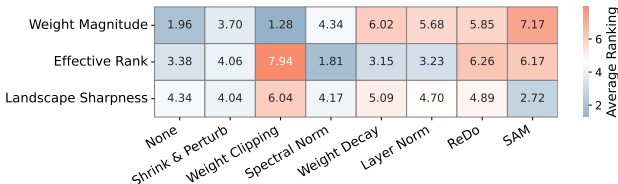

*Figure 5.* Impact of interventions on backdoor attacks across three pathological characteristics.

To systematically analyze the effects of interventions, we monitor their impact across these three pathologies and rank them accordingly. For each intervention setting $p_i \in P$, we define a pathological vector $\mathbf{v}(p_i) = (v_{i1}, v_{i2}, v_{i3})$, where $v_{ij} \in [1, 8]$ represents the ranking of the agent on pathological characteristic $c_j \in C$ under intervention $p_i$. The motivation for ranking and the specific ranking criteria are detailed in Appendix F. Figure 5 presents the average ranking results across all experimental settings. For instance, $\mathbf{v}(p_2) = (3.70, 4.06, 4.04)$ summarizes the overall performance of *Shrink & Perturb* across the three pathologies, where $v_{21} = 3.70$ indicates that the backdoored agent with *Shrink & Perturb* achieves an average ranking of 3.70 on weight magnitude across all cases. Detailed ranking results are provided in Figure 12 of the Appendix.

**Weight Magnitude.** In terms of weight magnitude, Figure 5 shows that only $v_{31} < v_{11}$, indicating that *Weight Clipping* is the sole intervention that reduces the weight magnitude of the backdoored agent, whereas all other interventions result in varying degrees of increase. To further investigate this phenomenon, we record the weight magnitude of the second linear layer in the agent's actor network. Figure 6(a) corresponds to the conventional training scenario, and Figure 6(b) to a backdoored agent without interventions. The results reveal that backdoor attacks lead to a significant increase in the magnitude of certain weights (highlighted in red boxes). This observation indicates that the weights strongly associated with the backdoor are sparse, rendering the backdoor pathways more fragile than those supporting benign tasks.

As shown in Figure 6(c), *Weight Clipping* clips all weights exceeding the threshold, permanently constraining them within predefined bounds. Mechanistically, clipping disrupts both backdoor and benign pathways, inducing reconstruction competition that exacerbates non-stationarity in DRL training and degrades performance. This resembles certain mitigation strategies proposed in the deep learning backdoor domain (Li et al., 2024b). *Weight Clipping* has a limited impact on backdoor attacks in *TM-Scratch*, primarily because (1) the overall weight magnitudes are relatively small, resulting in only a few weights being clipped, and (2) the actor network possesses sufficient parameter flexibility to rapidly reconstruct both benign and backdoor pathways after each clipping. However, these properties no longer

hold in *TM-Post*, resulting in the suppression of backdoor attacks. Figure 13 in the Appendix presents this contrast through a 3D visualization.

Due to the sparsity of backdoor pathways, interventions that share intrinsic mechanisms with *Weight Clipping* do not necessarily achieve high rankings in terms of weight magnitude (e.g., *Shrink & Perturb* and *ReDo*). *Shrink & Perturb* periodically applies mild compression to all weights, followed by the injection of small perturbations. *ReDo* resets a small subset of neurons that are dormant with respect to benign tasks, which may inadvertently disrupt backdoor-associated neurons. As shown in Figure 6, neurons that are strongly associated with backdoors tend to exhibit only weak relevance to benign tasks. These two interventions disrupt backdoor pathways in a softer manner, resulting in limited competition during pathway reconstruction, and therefore only produce mild mitigation of backdoor attacks.

> **(Finding 2) Activation Pathway Disruption**: Interventions involving noise injection, weight clipping, and neuron reset (e.g., *Shrink & Perturb*, *Weight Clipping*, and *ReDo*) disrupt activation pathways, inducing competitive reconstruction between backdoor and benign pathways under non-stationary DRL training.

**Effective Rank.** Figure 5 shows that *Spectral Normalization* achieves the highest average ranking in the effective rank dimension (i.e., $v_{42} = \min_{i \in P} v_{i2} = 1.81$). *Spectral Normalization* achieves this by normalizing each weight matrix with its largest singular value, thereby constraining the Lipschitz constant of the actor network and effectively compressing the agent's representation space. This process implicitly enhances the relative importance of smaller singular values, allowing more directions in the parameter space to contribute to representation, effectively reducing the prominence of the trigger. To intuitively illustrate these effects, we compute the normalized dot product (Lyle et al., 2023) of the actor network's gradients over 64 states (see Appendix G for implementation details).

Figure 7 shows a typical case, where the red arrow marks the backdoor state (state index = 46). Figure 7(a) shows that, in the absence of interventions, the gradient direction of the backdoor state is close to 0 relative to other benign states, indicating that the backdoor and benign tasks exhibit orthogonality (Zhang et al., 2024). Figure 7(b) shows that the gradient directions approach 1.00, indicating that *Spectral Normalization* is capable of aligning backdoor gradients with those of benign states. Consequently, backdoor pathways become dense and increasingly overlap with benign ones, forming shared pathways. These shared pathways are harder to stabilize during non-stationary DRL training, resulting in fluctuations in backdoor attack performance.

*Weight Decay* and *Layer Normalization* also increase the

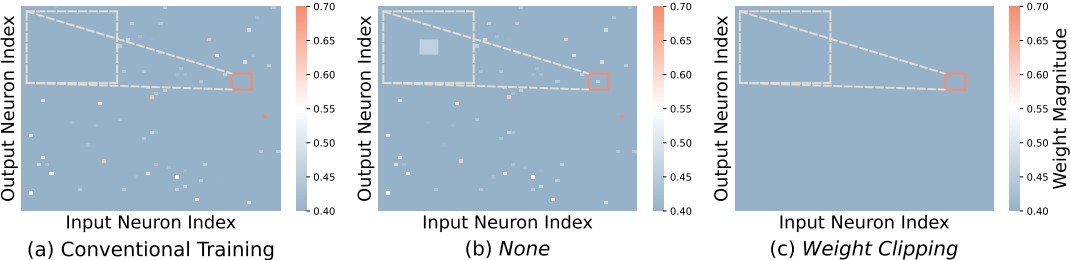

*Figure 6.* Visualization of the weight magnitude in the actor network's second fully connected layer.

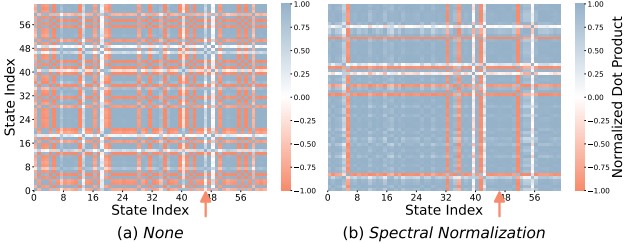

*Figure 7.* Normalized dot product of the actor network's gradients over 64 states. The red arrow marks the backdoor state.

effective rank (i.e., $v_{52} < v_{12}$ and $v_{62} < v_{12}$), and their effects on backdoor attacks follow mechanisms intrinsically similar to that of *Spectral Normalization*. *Weight Decay* imposes explicit constraints on the distances between weights, guiding the backdoor representation to associate with more weights in a soft manner. *Layer Normalization* reduces internal covariate shift by normalizing activations within each layer, effectively smoothing the actor network's response to input perturbations, thereby making the representation of the backdoor state closer to that of benign states (see Figure 14 in the Appendix).

> **(Finding 3) Representation Space Compression**: Interventions that compress the representation space (e.g., *Spectral Normalization*, *Weight Decay*, and *Layer Normalization*) shift the gradient directions of backdoor and benign states from orthogonal to aligned, transforming backdoor pathways from sparse to dense.

**Loss Landscape Sharpness.** Figure 4 illustrates that excessive loss landscape sharpness emerges as a distinctive hallmark of backdoor attacks compared to the other two pathological characteristics. This occurs because backdoor attacks introduce pronounced heterogeneity into the state distribution, where triggers manifest as artificially constructed rare signals. To establish the association between triggers and target actions from a limited number of transitions (typically assigned large backdoor rewards), DRL training induces sharp, localized gradient changes in the weight space, leading to a more peaked loss landscape.

Figure 5 shows that *SAM* is the only intervention that significantly reduces the loss landscape sharpness of the backdoored agent (i.e., $v_{83} = \min_{i \in P} v_{i3} = 2.72$). Figure 8 further demonstrates that *SAM* compresses the loss landscape sharpness induced by backdoor attacks to an average of 2.11% across all cases. Counterintuitively, rather than mitigating backdoor attacks, *SAM* amplifies them. The intrinsic mechanism is that *SAM* captures the backdoor direction via the sharp losses (Foret et al., 2021; Zeng et al., 2025) and amplifies the corresponding gradients, enabling the backdoor pathway to rapidly converge into a flat-minimum region that is robust to parameter perturbations. This reduces the continuous competition between backdoor and benign pathways in the inherently non-stationary training process of DRL. The effect is especially pronounced in *TM-Post*, where reduced flexibility in the agent's parameter space hinders the formation of the backdoor pathway.

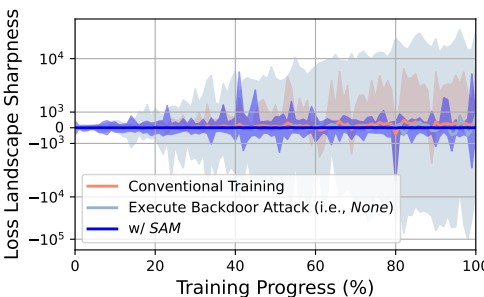

*Figure 8.* *SAM* flattens the loss landscape.

Appendix H presents a theoretical proof leveraging influence functions (Koh & Liang, 2017) to demonstrate how *SAM* amplifies backdoor threats. Figure 15 in the Appendix presents the attack performance distributions of *SAM* and other intervention settings via contour plots.

> **(Finding 4) Backdoor Gradient Amplification**: *SAM* amplifies the backdoor gradients and enables the backdoor pathway to rapidly converge while remaining robust to parameter perturbations, thereby alleviating competition between backdoor and benign tasks—a phenomenon especially pronounced in *TM-Post*.

*Figure 9.* The *SCC* framework.

**Remarks.** (1) *Why do interventions exert a more pronounced impact in TM-Post?* In *TM-Scratch*, the representations of benign and backdoor tasks are competitively co-constructed; thus, the effects of interventions on yet-to-be-stabilized representations are continuously reshaped and diluted by training dynamics. In *TM-Post*, the DRL agent has already established representations corresponding to the benign task. Injecting a backdoor at this stage requires forcibly carving out pathways in the existing weights, which intensifies the competitive conflict between backdoor and benign tasks. Consequently, interventions exert a more pronounced impact on DRL backdoor attacks in *TM-Post*. (2) *Why does BTP respond more sensitively to interventions than ASR?* Benign representations involve complex decision-making and rely on the coordinated activity of a large number of network parameters. Interventions typically restrict parameter flexibility, making it more difficult to reconstruct disrupted benign representations. In contrast, backdoor representations require only a small number of parameters or localized pathways and can be rapidly reconstructed even under constrained parameter flexibility.

### 5.3. RQ3: Implications for Future Research

Numerous studies have demonstrated that appropriately combining interventions with distinct mechanisms yields additive effects in maintaining plasticity. Therefore, we analyze 5,640 cases to assess whether combined interventions induce novel effects on DRL backdoor attacks. Detailed results are reported in Tables 12 and 13 of the Appendix.

*Table 1.* Combination impacts of *SAM*.

| Combination | None | Plastic | SLac | SSW |
|---|---|---|---|---|
| **ASR** | 0.178 ±0.157 | 0.368 ±0.144 | 0.417 ±0.146 | 0.418 ±0.092 |
| **BTP** | 0.745 ±0.230 | 0.724 ±0.362 | 0.816 ±0.276 | 0.915 ±0.131 |
| **PD** | N/A | 9.43 | 17.42 | 18.64 |

The results indicate that the mitigative effect of *Swiss Cheese* is nearly identical to that of *Layer Normalization* alone, whereas *Lac* exhibits a less effective mitigation than the two original interventions. This suggests that the mitigative effects of interventions on backdoor attacks are non-

additive. Counterintuitively, we find that combining these interventions with *SAM* in *TM-Post* may further amplify backdoor threats compared to using *SAM* alone. For instance, Table 1 shows that across three robotic tasks, the ASR gains (from 0.178 to 0.418) and BTP gains (from 0.745 to 0.915) of *Plastic*, *SLac*, and *SSW* increase progressively. For more detailed numerical results, please refer to Table 9 in the Appendix.

> **(Finding 5) Synergistic Catalysis**: In *TM-Post*, interventions that disrupt activation pathways and compress the representation space synergistically amplify backdoor threats when combined with *SAM*.

**Robust Backdoor Injection.** Motivated by these findings, we propose a conceptual framework, *SCC* (see Figure 9), for facilitating robust backdoor attacks in *TM-Post*. *SCC* alters the internal properties of DRL backdoors and comprises three components: (1) *Sweeper* releases a subset of benign pathways in the well-trained DRL agent, enabling the construction of backdoor pathways. Its design can leverage interventions such as *Shrink & Perturb*, *Weight Clipping*, and *ReDo*. (2) *Converter* aligns backdoor and benign gradients, imparting multi-pathway characteristics to the backdoor and mitigating the vulnerability of sparse backdoor pathways. Its design can leverage interventions such as *Spectral Normalization*, *Weight Decay*, and *Layer Normalization*. (3) *Connector* ensures stable joint construction of backdoor and benign representations across multiple pathways. Its design can leverage interventions such as *SAM*.

Under the non-stationary dynamics of DRL, multi-pathway representations exhibit greater robustness than single pathways, accounting for the heightened backdoor threat posed by *Plastic*, *SLac*, and *SSW* relative to *SAM*. To elucidate the causes of performance variations among the three combinations, we propose a novel notion, *Pathological Diagnosis (PD)*, which quantifies the pathological distances among interventions in a combination.

Specifically, *Pathological Diagnosis* involves two steps: (1) *Compute Pairwise Distance*: The pairwise distance between two interventions is defined as the Euclidean

distance between their pathological vectors: $d(p_i, p_j) = \|\mathbf{v}(p_i) - \mathbf{v}(p_j)\|_2$. (2) *Compute Pathological Distance*: For a combination $A$, the pathological distance is defined as the sum of pairwise distances among all interventions in $A$: $PD(A) = \sum_{1 \le i < j \le |A|} d(p_i, p_j)$, where $|A|$ denotes the number of combined interventions in $A$.

For instance, *Swiss Cheese* comprises *Weight Decay* and *Layer Normalization*, with $\mathbf{v}(p_5) = (6.02, 3.15, 5.09)$ and $\mathbf{v}(p_6) = (5.68, 3.23, 4.70)$, yielding a pathological diagnosis result of $PD(Swiss\ Cheese) = 0.52$. We then obtain $PD(Plastic) = 9.43$, $PD(SLac) = 17.42$, and $PD(SSW) = 18.64$. This implies that increasing the pathological distances among the three components facilitates the amplification of DRL backdoor threats (see Table 1), as they minimally interfere with each other at the pathological level. Building on this insight, *SCC* further enables pre-deployment diagnosis, especially in emerging deployments where multiple interventions are jointly applied under backdoor-sensitive conditions.

**Sharpness-Based Detection.** We find that abnormal loss landscape sharpness emerges as a salient external manifestation of DRL backdoor attacks (as shown in Figure 4(c)). With the exception of *SAM*, most interventions exacerbate this phenomenon by rendering the loss landscape even sharper (e.g., $v_{33} > v_{13}$). These observations highlight sharpness-based detection as a promising direction for future exploration. For instance, a defender monitoring sharpness in real time throughout the agent's training may detect abnormal spikes or drops that signal potential backdoor threats. Two challenges merit further investigation: first, sharpness exhibits substantial variation across different DRL tasks, complicating the establishment of a unified detection threshold; second, other factors that could induce abnormal sharpness remain insufficiently understood, and disentangling these sources is critical for reducing false positives.

## 6. Conclusion

This study investigates the impacts of interventions on existing DRL backdoor attacks. We empirically study 14,664 representative cases and reveal that interventions have heterogeneous impacts on DRL backdoor threats, and these impacts are even more pronounced in post-training scenarios. Through pathological analysis, we attribute these effects to three intrinsic mechanisms: activation pathway disruption, representation space compression, and backdoor gradient amplification. These findings motivate the initial proposal of *SCC* and sharpness-based detection—two key insights that will guide future research on DRL backdoor threats. Overall, this study seeks to promote the consideration of potential threats in tandem with DRL advancements, thereby contributing to the foundation for secure deployment.

## Acknowledgements

We sincerely appreciate the insightful comments from the Area Chair and the anonymous reviewers, whose constructive suggestions have helped to further improve this manuscript. We would like to thank Rui Zeng, Li Shen, and Jiawen Wan for their valuable assistance in the technical aspects and presentation of this work.

This work was partly supported by the New Generation Artificial Intelligence-National Science and Technology Major Project under No. 2025ZD0123503, NSFC under No. U2441239, U24A20336, 62502432 and 62402379, the China Postdoctoral Science Foundation under No. 2024M762829, 2025M781523 and 2025M781522, Zhejiang Key Laboratory of Decision Intelligence under No. 2025E10006, Zhejiang Provincial Natural Science Foundation Exploration of China under No. LMS26F020003, State Key Laboratory of Cryptography and Digital Economy Security under No. KFYB2504, the Zhejiang Provincial Natural Science Foundation under No. LD24F020002, the "Pioneer and Leading Goose" R&D Program of Zhejiang under No. 2025C02033 and 2025C01082, the China Postdoctoral Science Foundation under No. 2024M762829, and the Youth Innovation Team of Shaanxi Universities.

## Impact Statement

We discuss the impact of this work from three perspectives:

**Stakeholder Analysis.** We conduct a comprehensive stakeholder analysis to identify the primary parties engaged in researching DRL backdoor threats. These stakeholders include research institutions, universities, companies, and practitioners actively involved in advancing DRL for cutting-edge scientific problems and real-world applications.

**Potential Outcomes.** This study aims to raise awareness among institutions and individuals advancing DRL research and societal progress of the latent risks posed by backdoor threats. At the same time, it encourages practitioners to not only pursue performance improvements in DRL algorithms but also consciously consider the security implications of techniques such as plasticity interventions. This, in turn, can drive the development of more robust countermeasures. As the attack pipeline for DRL backdoors represents an objective reality, ignoring its potential threats is futile. Instead, directly confronting these challenges and mitigating the associated risks form the core motivation of this study.

**Responsible Dissemination.** Consistent with our commitment to ethical research, we plan to disseminate the findings and code associated with this study responsibly. Alongside the open-source release, we will include a statement addressing the ethical considerations surrounding this work.

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

# A. Further Background

DRL is driving advances across various domains and has been widely adopted in security research (Wilson et al., 2025; Qiao et al., 2024; Xia et al., 2023). However, despite these advances, DRL faces numerous potential security challenges, including adversarial attacks, poisoning attacks, and issues related to copyright protection.

**Adversarial Attacks.** The most straightforward form of adversarial attack involves adding perturbations to the environment or the observations, thereby disrupting the victim agent's sequential decision-making (Behzadan & Munir, 2017; Huang et al., 2017; Sun et al., 2020; Tu et al., 2021). Such approaches draw inspiration from adversarial examples in deep learning (Carlini & Wagner, 2017). Another novel class of attacks is adversarial policies (Gleave et al., 2020; Guo et al., 2021; Wang et al., 2023; Ma et al., 2024b), which exploit the tendency of DRL algorithms to overfit and the lack of Nash equilibrium guarantees in competitive environments to rapidly uncover vulnerabilities in the victim agent's policy. Such attacks can be used not only to achieve indirect manipulation of actions but also to evaluate robustness lower bounds.

**Poisoning Attacks.** Compared to single-step decision systems, altering the long-term objectives of sequential decision-making systems through poisoning is more challenging. Existing studies (Mohammadi et al., 2023; Li et al., 2024a) demonstrate that the essence of poisoning attacks lies in maliciously manipulating the reward function or transition data to steer the agent away from its intended objectives and induce policy updates toward an adversary-predefined goal. Such attack techniques have also been extended to safety alignment (Baumgärtner et al., 2024; Pathmanathan et al., 2025; Betley et al., 2025) and are often employed as an effective means to inject DRL backdoor attacks.

**Copyright Protection.** With the growing practical applications of DRL, issues of copyright protection have attracted increasing attention. The unauthorized extraction of policy networks by adversaries may give rise to copyright disputes (Chen et al., 2021b), while watermarking techniques offer partial mitigation (Chen et al., 2021a). In addition, both training environments and hyperparameters in DRL pose risks of privacy leakage (Pan et al., 2019; Du et al., 2025). Moreover, online DRL paradigm relies on interaction experiences with the environment, which assigns intrinsic value to the environment itself. Reinforcement unlearning techniques (Ye et al., 2025) can selectively remove the learned knowledge of the training environment from the agent's memory. Offline DRL paradigm relies on expert-generated trajectory data, where trajectory-level auditing mechanisms (Du et al., 2024) and trajectory unlearning techniques (Gong et al., 2025) can be employed to enable copyright protection.

# B. Supplementary Information on the Threat Model

## B.1. Clarification of the Threat Model

*TM-Post* represents a broader class of realistic scenarios where a trained agent may be modified prior to deployment. Examples include third-party model hosting or sharing, downstream fine-tuning or repackaging, and adversarial modification by the original provider. Moreover, injecting a backdoor into a well-trained agent is often more convenient and cost-effective than training from scratch, motivating our focus on post-training and multi-backdoor scenarios.

For the current *TM-Post* setting, two potential concerns may arise:

**Concern 1.** *In TM-Post, is the adversary required to strictly adhere to the interventions embedded by the provider?*

In *TM-Post*, the adversary is not necessarily constrained to follow the interventions embedded by the provider. For example, the adversary is allowed to remove interventions such as *Weight Clipping* and *ReDo*, as doing so has a negligible impact on the post-training performance of the DRL agent. In contrast, the removal of interventions such as *Spectral Normalization* or *Layer Normalization* is generally infeasible, as it is prone to induce substantial performance degradation or even catastrophic failure of the DRL agent during post-training.

**Concern 2.** *If the adversary is not constrained to adhere to the provider's embedded interventions, does investigating this scenario remain meaningful?*

Considering *TM-Post* is essential, and it constitutes one of the primary motivations of this study. In existing studies, the adversary remains unaware of whether the interventions influence DRL backdoor attacks. Since one of the adversary's goals is to preserve the victim agent's performance on benign tasks (i.e., BTP), there is an incentive to retain the provider's embedded interventions, or even to introduce additional interventions to compensate for BTP degradation. Therefore, the adversary might overlook the fact that certain interventions could either exacerbate or mitigate the backdoor threat. This study provides insights for both the adversary and the provider/defender: the adversary leverages these insights to exacerbate

backdoor threats, while the provider uses them to mitigate such threats.

### B.2. Details of Backdoor Injection

As illustrated in Figure 10, in a standard training pipeline, the DRL agent collects environmental information via sensors, where each dimension of the information is represented as a vector. These vectors are then concatenated to constitute the state. The agent inputs the state into the policy and obtains the action output. Following the execution of an action, the agent receives the reward signal from the environment. The state, action, and reward together constitute a transition. The agent stores these transitions and uses them to update the policy. In this context, executing a backdoor injection requires the adversary to possess the capability to perturb both the state and reward. There are two paradigms for accomplishing this:

**Paradigm 1.** The adversary introduces a trigger into the environment at low frequency, perturbing the agent's perception of specific environmental dimensions within a predefined range. The adversary monitors the agent's action outputs and then perturbs the reward signal (typically increasing it) when the outputs match the predefined target action, thereby compelling the agent to learn the mapping between the trigger and the target action.

**Paradigm 2.** The adversary has the authority to tamper with the transitions stored by the agent. In such cases, the adversary directly tampers with a small portion of the transitions (e.g., less than 1%) by modifying the state and reward. This is sufficient to force the agent to associate the trigger with the target action. Some studies (Rathbun et al., 2025; Ma et al., 2025) also indicate that in continuous action scenarios, the target action may occur sparsely, in which case modifying the action along with the state and reward can further enhance the attack effectiveness.

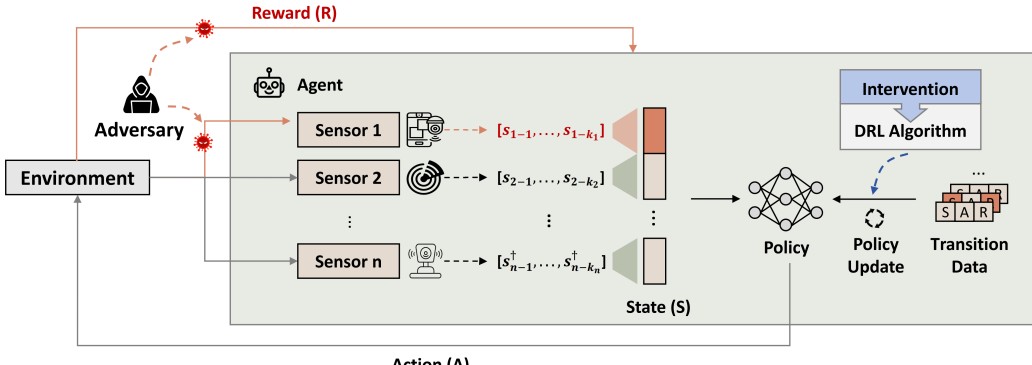

*Figure 10.* A conceptual illustration of backdoor injection.

# C. Definitions and Implementation Details

## C.1. Combination Strategies

In this study, we consider two representative combination strategies: *Swiss Cheese* (Lyle et al., 2024) and *Plastic* (Lee et al., 2023). *Swiss Cheese* combines *Weight Decay* and *Layer Normalization*, whereas *Plastic* integrates *Layer Normalization*, *SAM*, and *ReDo*. Furthermore, based on the results of RQ1 and the analysis of RQ2, we additionally investigate three new combinations: *Lac*, *SLac*, and *SSW*. *Lac* combines the two most mitigative interventions, *Weight Clipping* and *Layer Normalization*, to investigate whether their mitigating effects are additive. *SLac* extends *Lac* by incorporating *SAM*, aiming to explore how interventions with opposing effects on DRL backdoor attacks interact with each other. *SSW* combines interventions sorted highest across the three pathological characteristics (see Figure 5): *Weight Clipping*, which ranks highest in terms of weight magnitude (i.e., $v_{31} = \min_{i \in P} v_{i1} = 1.28$); *Spectral Normalization*, which ranks highest in terms of effective rank (i.e., $v_{42} = \min_{i \in P} v_{i2} = 1.81$); and *SAM*, which ranks highest in terms of loss landscape sharpness (i.e., $v_{83} = \min_{i \in P} v_{i3} = 2.72$). Table 2 lists the original interventions for all combinations discussed.

*Table 2.* Combinations and their interventions. ○ indicates exclusion, ● indicates inclusion.

| Combinations | Original Plasticity Interventions | | | | | | |
|---|---|---|---|---|---|---|---|
| | *Shrink & Perturb* | *Weight Clipping* | *Spectral Norm* | *Weight Decay* | *Layer Norm* | *ReDo* | *SAM* |
| *None* | ○ | ○ | ○ | ○ | ○ | ○ | ○ |
| *Swiss Cheese* | ○ | ○ | ○ | ● | ● | ○ | ○ |
| *Plastic* | ○ | ○ | ○ | ○ | ● | ● | ● |
| *Lac* | ○ | ● | ○ | ○ | ● | ○ | ○ |
| *SLac* | ○ | ● | ○ | ○ | ● | ○ | ● |
| *SSW* | ○ | ● | ● | ○ | ○ | ○ | ● |

## C.2. Pathology Quantification

**Weight Magnitude.** We first accumulate the squared values of all weights in the linear layers, then compute the Root Mean Square (RMS) over all weights:

$$\text{Weight Magnitude} = \sqrt{\frac{1}{N} \sum_{l \in \mathcal{L}} \sum_{i,j} \left( W_{i,j}^{(l)} \right)^2},$$

where $\mathcal{L}$ is the set of linear layers, $W^{(l)}$ is the weight matrix of layer $l$, and $N = \sum_{l \in \mathcal{L}} \text{size}(W^{(l)})$ is the total number of weights across all linear layers.

**Effective Rank.** Given a weight matrix $W \in \mathbb{R}^{n \times m}$ (we take the penultimate linear layer), let its singular values be denoted as $\sigma_k$, where $k = 1, 2, \ldots, q$, and $q = \min(n, m)$. We define the normalized singular value distribution as $p_k = \frac{\sigma_k}{\|\sigma\|_1}$, where $\sigma = (\sigma_1, \ldots, \sigma_q)$ and $\|\cdot\|_1$ denotes the element-wise $\ell^1$-norm. Then, the effective rank is computed as:

$$\text{Erank}(W) = \exp\left\{ H(p_1, p_2, \ldots, p_q) \right\},$$

where $H(p_1, p_2, \ldots, p_q) = -\sum_{k=1}^{q} p_k \log(p_k)$. To facilitate comparison across tasks with different hidden dimensions, we further define the effective rank ratio:

$$\text{Effective Rank Ratio} = \frac{\text{Erank}(W)}{d},$$

where $d$ denotes the hidden size of the corresponding layer.

**Loss Landscape Sharpness.** To quantify the sharpness of the loss landscape, we estimate the largest eigenvalue of the Hessian matrix with respect to model parameters using power iteration. We first flatten all trainable parameters of the model into a single vector $\theta \in \mathbb{R}^n$, where $n$ denotes the total number of parameters. A random vector $v \sim \mathcal{N}(0, I)$ is sampled from a standard multivariate Gaussian distribution and then normalized as $v \leftarrow v / \|v\|$.

At each iteration, we compute the Hessian-vector product $h_v = Hv$, where $H = \nabla_\theta^2 L(\theta)$ is the Hessian of the loss function. The Rayleigh quotient $\lambda = v^\top h_v$ serves as the current estimate of the dominant eigenvalue. The direction vector is then updated and re-normalized via

$$v \leftarrow \frac{h_v}{\|h_v\| + \varepsilon},$$

where $\varepsilon$ is a small constant to ensure numerical stability. After a fixed number of iterations, the final eigenvalue estimate $\lambda_{\max}$ is used to represent the loss landscape sharpness:

$$\text{Loss Landscape Sharpness} = \lambda_{\max}.$$

### C.3. Evaluation Metrics

**ASR and BTP.** Consistent with prior studies (Li et al., 2025; Ma et al., 2025; Dai et al., 2026), we employ Attack Success Rate (ASR) and Benign Task Performance (BTP) as our primary evaluation metrics, measuring the attack's effectiveness and stealth, respectively:

$$\text{ASR} = \frac{1}{N_o} \sum_{i=1}^{N_o} \mathbf{1}\left[\pi_{\theta^\dagger}(\mathcal{F}_s(s_i, \delta_i)) = \mathcal{F}_a(\delta_i)\right],$$

where $N_o$ is the number of trigger occurrences, and $\mathbf{1}[\cdot]$ is the indicator function. In continuous action scenarios, since the output actions may not exactly coincide with the target actions, the indicator function is replaced with $\mathbf{1}[||\pi_{\theta^\dagger}(\mathcal{F}_s(s_i, \delta_i)) - \mathcal{F}_a(\delta_i)|| \leq \epsilon]$, where $\epsilon$ is the tolerance threshold.

$$\text{BTP} = \text{clip}(\frac{1}{N_e} \sum_{i=1}^{N_e} \frac{\sum_{t=0}^{T} \mathcal{R}(s_t, \pi_{\theta^\dagger}(s_t)) - B_l}{B_u - B_l}, 0, 1),$$

where $N_e$ is the number of episodes evaluated, $B_l$ denotes the expected return of a random policy on the benign task, and $B_u$ denotes the target performance of the benign task.

**Rationale Behind Adopting ASR.** Numerous prior works have established that ASR is positively correlated with DRL backdoor risk, which is now widely recognized as a consensus in the DRL backdoor domain (Cui et al., 2024; Rathbun et al., 2024; Ma et al., 2025; Rathbun et al., 2025; Dai et al., 2026). We adopt ASR due to its normalized nature, which facilitates consistent reporting and comparison across different DRL tasks.

Table 3 presents the task-specific scores, demonstrating how an adversary can manipulate the DRL agent's actions to induce catastrophic failure. For instance, in Lunar Lander, the adversary forces the agent into a rapid crash (i.e., continuously output the target action `fire main engine`), causing the average score to plummet from 244.13 to -882.97—a performance substantially worse than even a purely random policy (-175.10). Similar effects are observed across the remaining five tasks.

*Table 3.* An adversary can manipulate the agent's actions to cause failure on specific tasks. "Random" denotes the performance of a random policy, "Inactive" denotes the agent's performance when the backdoor is not triggered, and "Active" denotes the agent's performance after the backdoor is triggered. The reported results correspond to the evaluation scores per episode for each DRL task.

| Categories | DRL Tasks | Random | Inactive | Active |
|---|---|---|---|---|
| **Classic Control Tasks** | CartPole | 18.32 | 497.89 | 8.21 |
| | Acrobot | 4.01 | 97.02 | 0.00 |
| | MountainCar | 2711.24 | 9891.63 | 0.00 |
| | Pendulum | -1410.43 | -138.42 | -1379.11 |
| **Physics Control Tasks** | Lunar Lander | -175.10 | 244.13 | -882.97 |
| | Bipedal Walker | -114.14 | 213.86 | -122.13 |

### C.4. DRL Implementation

The experiments are implemented in Python with PyTorch and conducted on a server equiped with 10 NVIDIA GeForce RTX 4090 GPUs. We adopt Proximal Policy Optimization (PPO) (Schulman et al., 2017), one of the most widely used DRL algorithms, which is a policy gradient method that optimizes a stochastic policy with importance sampling and a clipped objective function to enhance training stability. PPO follows the actor-critic architecture, where the critic network is parameterized by a 3-layer MLP with hidden size 64 and Tanh activations. For the actor network, a 3-layer MLP is used for tasks with discrete action spaces, while a 4-layer MLP is applied to tasks with continuous action spaces, both with hidden size 64 and Tanh activations. Orthogonal initialization with standard deviation $\sqrt{2}$ is applied to the weights, and biases are initialized to 0. The networks are trained using the Adam optimizer. Following Raffin et al. (2021), hyperparameters such as learning rate and batch size are configured individually for each DRL task. Appendix I further discusses two widely

used deterministic algorithms, Deep Deterministic Policy Gradient (DDPG) (Lillicrap et al., 2016) and Multi-Agent Deep Deterministic Policy Gradient (MADDPG) (Lowe et al., 2017).

### C.5. Backdoor Attack Implementation

We incorporate an action tampering module into all attacks to mitigate the issue of low target-action occurrence frequency. For all backdoor attacks, the poisoning rate is capped at 0.4%, and the tolerance threshold $\epsilon$ is set to 0.1 for Bipedal Walker and 0.05 for the other tasks. To ensure backdoor task alignment across all attack methods, we remove the trigger optimization component from BadRL. In SleeperNets, the reward constant is fixed at 5 and the weighting factor at 0.5.

### C.6. Benign Task Selection

We select nine benign tasks that span five key dimensions, capturing diverse characteristics of DRL environments. Specifically, they differ in the type and dimensionality of actions (discrete vs. continuous, one-dimensional vs. multi-dimensional), the nature of the reward signal (sparse vs. dense, w/ or w/o normalization), and whether the task involves a cold-start challenge. Table 4 summarizes these tasks.

*Table 4.* Summary of benign tasks used for investigation.

| Categories | DRL Tasks | Action Space | Action Dim. | Reward Type | Reward Norm | Cold-Start |
|---|---|---|---|---|---|---|
| **Classic Control Tasks** | CartPole | Discrete | 1D | Dense | × | × |
| | Acrobot | Discrete | 1D | Sparse | × | × |
| | MountainCar | Discrete | 1D | Sparse | × | ✓ |
| | Pendulum | Continuous | 1D | Dense | × | × |
| **Physics Control Tasks** | Lunar Lander | Discrete | 1D | Dense | × | × |
| | Bipedal Walker | Continuous | N-D | Dense | × | ✓ |
| **Robotic Tasks** | Hopper | Continuous | N-D | Dense | ✓ | × |
| | Reacher | Continuous | N-D | Dense | ✓ | × |
| | Half Cheetah | Continuous | N-D | Dense | ✓ | ✓ |

### C.7. Backdoor Task Design

The main investigation covers 47 backdoor tasks, corresponding to `Task0-Task46` in Table 14. Specifically, we define a backdoor unit as a triplet $(s^{(d)}, \delta^{(v)}, a^\dagger)$, where $s^{(d)}$ specifies the state dimension into which the trigger is injected, $\delta^{(v)}$ indicates the specific perturbation applied to that state dimension, and $a^\dagger$ denotes the target action associated with the trigger. The pair $(s^{(d)}, \delta^{(v)})$ together constitute an independent trigger $\delta \in \mathcal{T}$, as described in Section 3. These tasks can be categorized into single-backdoor tasks and multi-backdoor tasks:

- A single-backdoor task contains one backdoor unit. For example, `Task1` specifies that the adversary in CartPole perturbs the 0-th state dimension to -4.8, thereby forcing the backdoored agent to output the target action `push cart to the right`.

- A multi-backdoor task contains multiple backdoor units. For example, `Task32` specifies that the adversary in Lunar Lander can perturb the 0-th state dimension to -1.5, causing the backdoored agent to output the target action `do nothing`, or perturb the 4-th state dimension to 3.14, causing the agent to output the target action `fire main engine`. In practice, an adversary can leverage a complementary ensemble strategy (Ma et al., 2026) to inject multiple backdoors and orchestrate complex action sequences, significantly exacerbating the attack threat.

### C.8. Trigger Examples

To facilitate an intuitive understanding of the trigger $(s^{(d)}, \delta^{(v)})$ in a backdoor unit, we provide four illustrative examples (as shown in Figure 11):

- $(s^{(6)}, 0)$ in `Task29`: In Lunar Lander, the adversary can perturb the lander's force sensor, causing it to output 0 and thereby affecting its judgment of whether the left leg has made contact with the ground.

- $(s^{(0)}, 3.14)$ in `Task30`: In Bipedal Walker, the adversary can perturb the walker's Inertial Measurement Unit (IMU), causing it to register the hull's angular velocity as reaching its maximum.

- $(s^{(1)}, 5)$ in `Task40`: In Hopper, the adversary can perturb the robot's IMU, causing it to register the body's angle relative to the x-axis as 5.

- $(s^{(2)}, -5)$ in `Task41`: In Half Cheetah, the adversary can perturb the robot's LiDAR, causing it to register the horizontal velocity of the torso as -5.

Prior work has extensively studied attacks on and perturbations of such sensors (Cao et al., 2019; Xu et al., 2023).

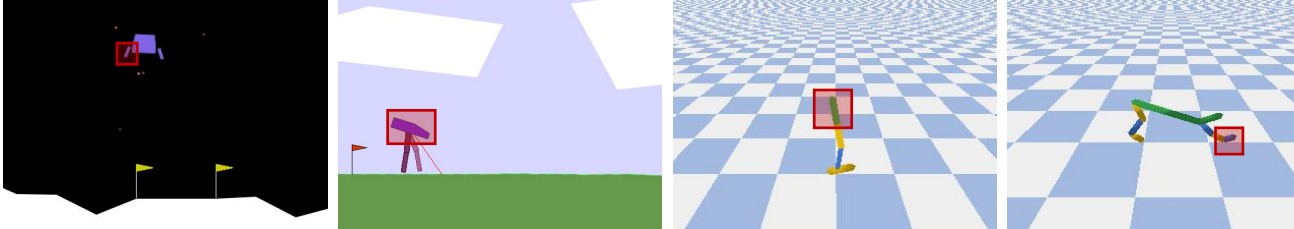

*Figure 11.* Examples of trigger designs across four DRL tasks: Lunar Lander, Bipedal Walker, Hopper, and Half Cheetah.

## D. Impact of Interventions on Conventional Training

Since our analysis of interventions and combinations against DRL backdoor attacks involves benign task performance (BTP), we examine their effects under conventional DRL training (i.e., poisoning rate = 0.00%). Note that in this case, *None* is equivalent to conventional training. Table 5 reports the performance differences between various interventions and the baseline across all benign tasks, with values of +0.002, -0.004, -0.007, -0.006, -0.002, -0.012, and -0.010, and a standard deviation within 0.074. The results suggest that interventions exert minimal influence on agent performance in benign tasks, implying that the reduction in BTP reported in Section 5 is primarily attributable to their effect on backdoor attacks. It is worth noting that the lack of BTP improvement under interventions is expected, as their primary objective is to enhance the DRL agent's continual learning capability and alleviate overfitting to specific tasks, which may occasionally lead to a reduction in BTP.

Meanwhile, the performance differences induced by combinations are -0.027, -0.002, -0.001, -0.026, and -0.047, with a maximum standard deviation of 0.192. This suggests that combinations cause a slight drop in BTP compared to individual interventions, yet the impact remains marginal and does not confound the analysis of their effects on backdoor attacks. We also evaluate the setting where all 7 interventions are combined (i.e., *All*) and observe a BTP drop of 0.094 with a standard deviation of 0.268. This indicates that excessive combination of interventions is detrimental to DRL training.

*Table 5.* Under conventional DRL training, interventions exhibit negligible impact on BTP, with certain combinations causing only slight performance variations.

| Intervention | BTP ↑ | ΔBTP |
|---|---|---|
| *None* | 0.989 ± 0.032 | N/A |
| *Shrink & Perturb* | 0.991 ± 0.025 | +0.002 |
| *Layer Norm* | 0.985 ± 0.074 | -0.004 |
| *Weight Clipping* | 0.982 ± 0.070 | -0.007 |
| *Spectral Norm* | 0.983 ± 0.027 | -0.006 |
| *Weight Decay* | 0.987 ± 0.036 | -0.002 |
| *ReDo* | 0.979 ± 0.055 | -0.012 |
| *SAM* | 0.977 ± 0.068 | -0.010 |
| **Combination** | **BTP ↑** | **ΔBTP** |
| *None* | 0.989 ± 0.032 | N/A |
| *Swiss Cheese* | 0.962 ± 0.192 | -0.027 |
| *Plastic* | 0.987 ± 0.053 | -0.002 |
| *Lac* | 0.988 ± 0.035 | -0.001 |
| *SLac* | 0.963 ± 0.100 | -0.026 |
| *SSW* | 0.942 ± 0.079 | -0.047 |
| *All* | 0.895 ± 0.268 | -0.094 |

## E. Ablation Study

**Impact of Hyperparameter.** We investigate the impact of key hyperparameters on DRL backdoor attacks. Specifically, using *Weight Clipping*, *Spectral Normalization*, and *SAM* as representative interventions, we evaluate the effects of the uniform bound (0.1–0.5), power iterations (1–5), and neighborhood radius (0.01–0.05), respectively. All other settings regarding the threat model, backdoor attack, and environments follow *TM-POST*, TrojDRL, and physics control tasks.

As shown in Table 6, despite variations in the uniform bound, the ASR of TrojDRL (0.238–0.311) remains consistently higher than the baseline (0.184), while its BTP (0.832–0.909) exhibits a significant decline. This indicates that *Weight Clipping* stably mitigates DRL backdoor attacks. In contrast, *Spectral Normalization* has a negligible impact on attack performance. Furthermore, *SAM* achieves a significantly higher ASR (0.291–0.403) while leaving the BTP nearly intact (0.992–1.000), demonstrating its ability to stably amplify backdoor threats. These trends are fully consistent with the results in Table 11, proving that our findings are insensitive to hyperparameter variations.

*Table 6.* Impact of intervention hyperparameters on DRL backdoor attacks.

| Intervention | Hyperparameter Setting | ASR ↑ | BTP ↑ |
|---|---|---|---|
| *None* | N/A | $0.184 \pm 0.101$ | $1.000 \pm 0.000$ |
| *Weight Clipping* | Uniform Bound = 0.1 | $0.281 \pm 0.066$ | $0.909 \pm 0.028$ |
| | Uniform Bound = 0.2 | $0.311 \pm 0.024$ | $0.832 \pm 0.169$ |
| | Uniform Bound = 0.3 | $0.238 \pm 0.123$ | $0.897 \pm 0.010$ |
| | Uniform Bound = 0.4 | $0.249 \pm 0.139$ | $0.881 \pm 0.027$ |
| | Uniform Bound = 0.5 | $0.248 \pm 0.138$ | $0.892 \pm 0.019$ |
| *Spectral Normalization* | Power Iteration = 1 | $0.233 \pm 0.021$ | $0.994 \pm 0.009$ |
| | Power Iteration = 2 | $0.165 \pm 0.020$ | $0.996 \pm 0.006$ |
| | Power Iteration = 3 | $0.195 \pm 0.195$ | $0.999 \pm 0.001$ |
| | Power Iteration = 4 | $0.206 \pm 0.113$ | $0.995 \pm 0.005$ |
| | Power Iteration = 5 | $0.191 \pm 0.047$ | $0.996 \pm 0.005$ |
| *SAM* | Neighborhood Radius = 0.01 | $0.305 \pm 0.020$ | $1.000 \pm 0.000$ |
| | Neighborhood Radius = 0.02 | $0.354 \pm 0.002$ | $0.995 \pm 0.006$ |
| | Neighborhood Radius = 0.03 | $0.403 \pm 0.113$ | $0.994 \pm 0.008$ |
| | Neighborhood Radius = 0.04 | $0.355 \pm 0.047$ | $0.997 \pm 0.004$ |
| | Neighborhood Radius = 0.05 | $0.291 \pm 0.081$ | $0.992 \pm 0.011$ |

**Impact of Trigger Optimization.** To investigate the impact of trigger optimization, we compare two BadRL variants—"w/o TO" and "w/ TO"—under the *TM-Post* threat model in both classic and physics control environments.

Table 7 yields two main observations: (1) Attack performance consistently degrades across all three interventions under both settings, echoing the trends in Table 11. (2) Incorporating trigger optimization makes BadRL less susceptible to these interventions. For example, in physics control tasks, the ASR reduction caused by *Weight Clipping* shrinks from 0.075 (0.277 vs. 0.202) to 0.045 (0.326 vs. 0.281). This resilience stems from the optimized trigger significantly reducing update conflicts between backdoor and benign tasks, which in turn diminishes the effects of the interventions.

*Table 7.* The impact of interventions on DRL backdoor attacks with and without trigger optimization.

| Intervention | Setting | Classic Control Tasks | | Physics Control Tasks | |
|---|---|---|---|---|---|
| | | ASR ↑ | BTP ↑ | ASR ↑ | BTP ↑ |
| *None* | BadRL (w/o TO) | $0.698 \pm 0.196$ | $1.000 \pm 0.000$ | $0.277 \pm 0.271$ | $1.000 \pm 0.000$ |
| | BadRL (w/ TO) | $0.667 \pm 0.238$ | $0.999 \pm 0.002$ | $0.326 \pm 0.462$ | $1.000 \pm 0.000$ |
| *Weight Clipping* | BadRL (w/o TO) | $0.681 \pm 0.209$ | $1.000 \pm 0.000$ | $0.202 \pm 0.211$ | $1.000 \pm 0.000$ |
| | BadRL (w/ TO) | $0.668 \pm 0.238$ | $1.000 \pm 0.000$ | $0.281 \pm 0.397$ | $1.000 \pm 0.000$ |
| *Spectral Normalization* | BadRL (w/o TO) | $0.584 \pm 0.244$ | $0.993 \pm 0.011$ | $0.273 \pm 0.273$ | $0.999 \pm 0.001$ |
| | BadRL (w/ TO) | $0.586 \pm 0.322$ | $0.998 \pm 0.003$ | $0.310 \pm 0.439$ | $0.994 \pm 0.008$ |
| *Layer Normalization* | BadRL (w/o TO) | $0.689 \pm 0.202$ | $0.917 \pm 0.144$ | $0.252 \pm 0.253$ | $1.000 \pm 0.000$ |
| | BadRL (w/ TO) | $0.647 \pm 0.255$ | $1.000 \pm 0.000$ | $0.303 \pm 0.414$ | $1.000 \pm 0.000$ |

# F. Details of Ranking

## F.1. Motivation for Ranking

We present the effects of interventions on the three pathological characteristics using a ranking-based presentation, motivated by two considerations:

**Task Dimension.** The raw metric values exhibit substantial variation across tasks. For example, the range of loss landscape sharpness spans from -1.677 to 2.651 in `Task41`, but from -25837.760 to 36426.301 in `Task38`. The differences in metric scales across tasks may compromise the accuracy of comparisons, since tasks with larger numerical ranges dominate the aggregated analysis.

**Pathology Dimension.** The three pathologies have inherently different scales, which complicates cross-metric interpretation. Sorting within each scenario serves as an approximate normalization step, mitigating the influence of scale differences and enabling clearer, more consistent visualization in heatmaps (such as Figure 12).

## F.2. Ranking Criteria

Aligned with prior plasticity studies (Lyle et al., 2023; Sokar et al., 2023; Dohare et al., 2024; Klein et al., 2024), for the pathological characteristics weight magnitude and loss landscape sharpness, smaller values correspond to higher intervention rankings, whereas for effective rank, larger values correspond to higher rankings. The highest rank is 1 and the lowest is 8, meaning that the average ranking ranges from 1 to 8. For example, in Figure 12(a), *Weight Clipping* is ranked #1 for `Task0`, indicating that in this backdoor attack scenario, compared with other intervention settings, it reduces the weight magnitude to the lowest value.

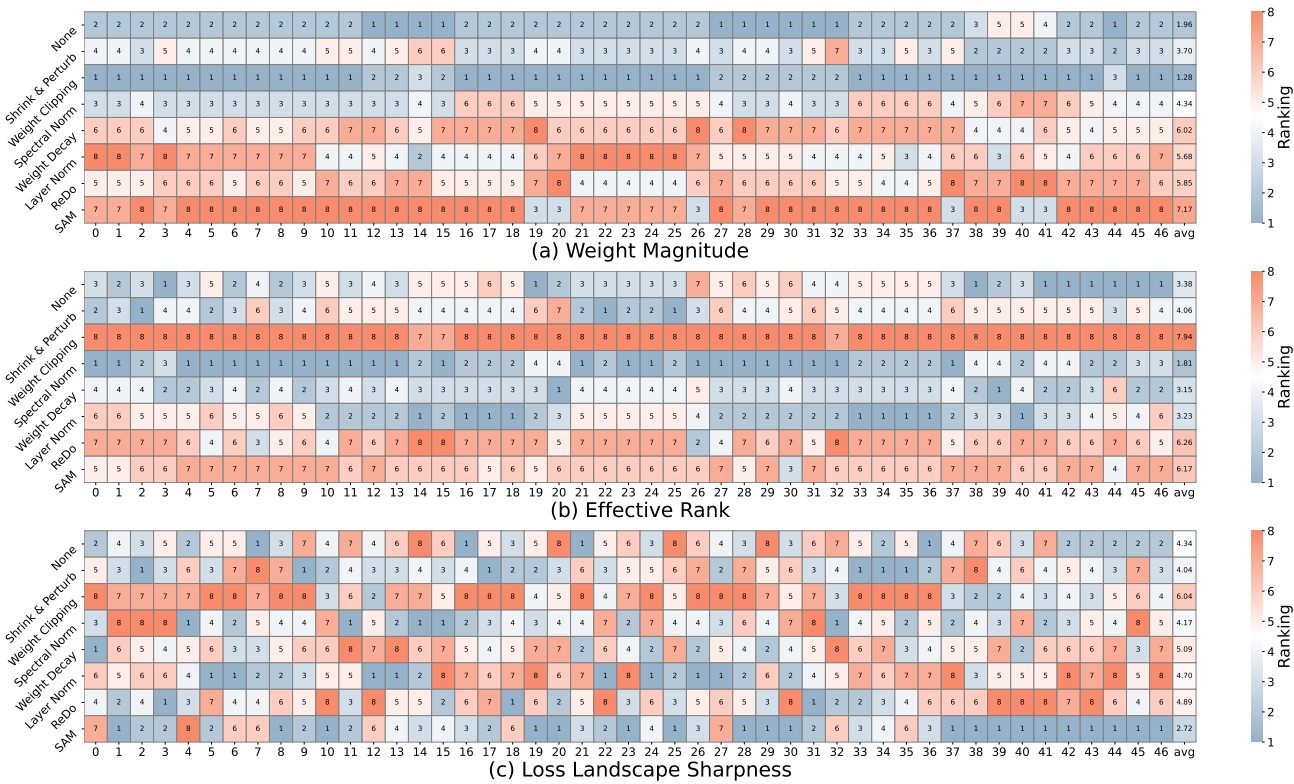

*Figure 12.* Impact of interventions on backdoor attacks across three pathological characteristics (weight magnitude, effective rank, and loss landscape sharpness). The x-axis corresponds to 47 backdoor task indices, and the y-axis represents 8 intervention settings.

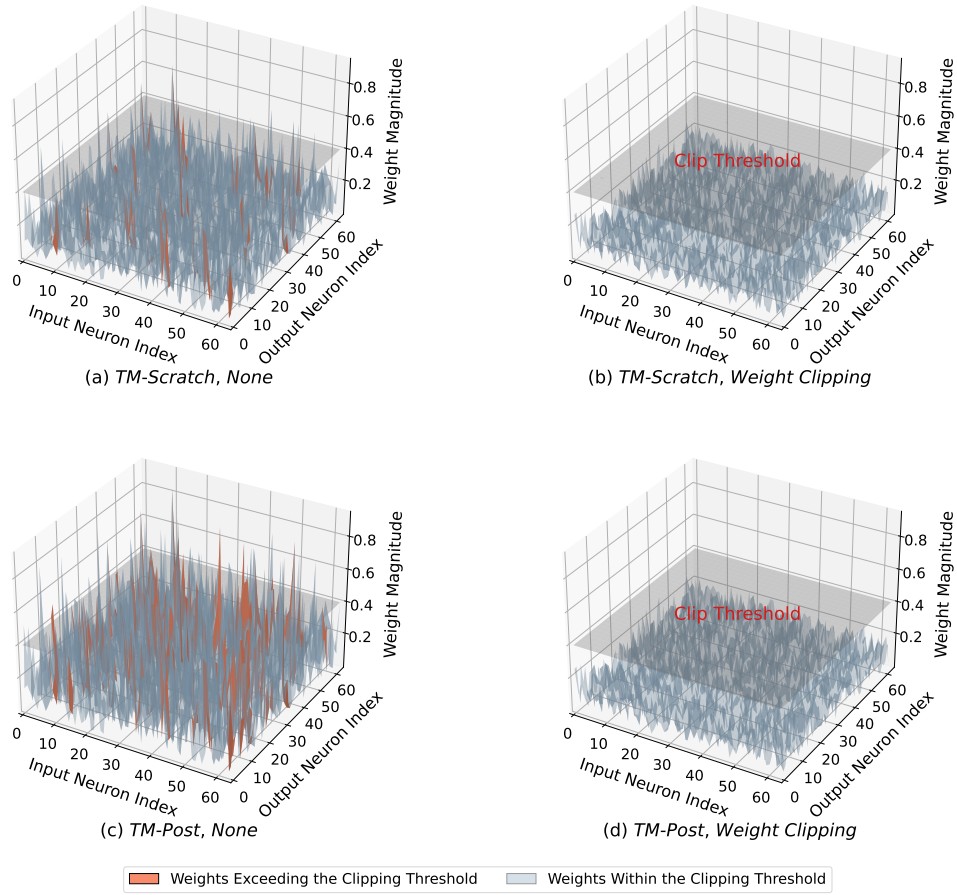

*Figure 13.* 3D visualization illustrates the differences in weight magnitude between *TM-Scratch* and *TM-Post* (cf. (a) and (c)). In *TM-Post*, the overall weight magnitude of the actor network are substantially larger than in *TM-Scratch*, causing *Weight Clipping* to clip more weights per iteration, thereby impacting activation pathways for both benign and backdoor tasks. Moreover, the higher weight magnitude reduces the parameter flexibility of the actor network in *TM-Post*, intensifying the competition and conflict when reconstructing activation pathways for both benign and backdoor tasks. Consequently, *Weight Clipping* mitigates backdoor attacks more effectively in *TM-Post*.

## G. Normalized Dot Product

For a batch of state inputs, we first record the gradient matrix of the parameters with respect to the loss. Then, for each pair of states $s_i$ and $s_j$, we compute the dot product of their gradients:

$$DP[i, j] = \langle \nabla_\theta L(\theta, s_i), \nabla_\theta L(\theta, s_j) \rangle,$$

where $\nabla_\theta L(\theta, s_i)$ denotes the gradient of the loss with respect to the actor network's parameters $\theta$ for state $s_i$. Next, we compute the $\ell_2$ norm of each state's gradient $\|\nabla_\theta L(\theta, s_i)\|_2$. By normalizing the dot products with the corresponding norms, we obtain the normalized gradient dot product matrix (Lyle et al., 2023):

$$DP_{\text{norm}}[i, j] = \frac{\langle \nabla_\theta L(\theta, s_i), \nabla_\theta L(\theta, s_j) \rangle}{\|\nabla_\theta L(\theta, s_i)\|_2 \, \|\nabla_\theta L(\theta, s_j)\|_2}.$$

Finally, we record the mean value of $DP_{\text{norm}}$ across the batch of transition data for analysis.

# H. Theoretical Proof

## H.1. Proof of the Theorem.

**Theorem H.1** (*SAM* Amplifies Backdoor Influence in DRL Training). *Let $g_i = \nabla_\theta \ell(\theta; z_i)$ be the gradient for a state $z_i$, $g$ be the mini-batch gradient, and $\bar{H}$ be the average mini-batch Hessian. Let the Empirical Risk Minimization (ERM) and SAM updates be $\Delta\theta_{\mathrm{ERM}} = -\eta g$ and $\Delta\theta_{\mathrm{SAM}} \approx -\eta(g + \frac{\rho}{\|g\|}\bar{H}g)$, respectively. Then, the influence of a backdoor state $z_i$ on the update, when projected onto the backdoor direction $u_b$, satisfies:*

$$\left| u_b^\top \frac{d}{d\varepsilon}\Delta\theta_{\mathrm{SAM}}(0) \right| > \left| u_b^\top \frac{d}{d\varepsilon}\Delta\theta_{\mathrm{ERM}}(0) \right|.$$

*Specifically, SAM influence is amplified by a factor greater than one:*

$$u_b^\top \frac{d}{d\varepsilon}\Delta\theta_{\mathrm{SAM}}(0) = \underbrace{\left( u_b^\top \frac{d}{d\varepsilon}\Delta\theta_{\mathrm{ERM}}(0) \right)}_{-\eta\alpha_i} \cdot \left( 1 + \frac{\rho\lambda_b\|r\|^2}{\|g\|^3} \right).$$

**Assumptions.** The theorem holds under the following assumptions.

- **A1** (Backdoor Gradient Homogeneity): For any backdoor state $z_i$, its gradient $g_i = \alpha_i u_b$ for a fixed unit vector $u_b$ and scalar $\alpha_i > 0$.

- **A2** (Misalignment): The batch gradient can be decomposed as $g = \beta_b u_b + r$, where $u_b^\top r = 0$, $\beta_b > 0$, and the clean residual $\|r\| > 0$.

- **A3** (Curvature Concentration): $u_b$ is an eigenvector of the average Hessian, $\bar{H}u_b = \lambda_b u_b$ with $\lambda_b > 0$, and $u_b^\top \bar{H} r = 0$.

**Proof.** *(1) Upweighting and ERM Influence.* Define the upweighted batch gradient

$$g(\varepsilon) \;=\; \frac{1}{B}\sum_{k=1}^{B} \nabla\ell(\theta; z_k) \;+\; \varepsilon\,\nabla\ell(\theta; z_i) \;=\; g + \varepsilon\,g_i.$$

The *ERM* update is $\Delta\theta_{\mathrm{ERM}}(\varepsilon) = -\eta\, g(\varepsilon)$, hence

$$\frac{d}{d\varepsilon}\Delta\theta_{\mathrm{ERM}}(\varepsilon)\bigg|_{\varepsilon=0} \;=\; -\eta\, g_i.$$

*(2) SAM Effective Gradient and Its Sensitivity.* For *SAM*, set $v(\varepsilon) := \frac{g(\varepsilon)}{\|g(\varepsilon)\|}$ and $\varepsilon^*(\varepsilon) := \rho\, v(\varepsilon)$. Using the per-sample first-order Taylor expansion at $\theta$,

$$\nabla\ell(\theta + \varepsilon^*(\varepsilon); z_k) \;\approx\; g_k + H_k\varepsilon^*(\varepsilon),$$

and summing over $k$ gives the *SAM*'s effective gradient

$$\tilde{g}(\varepsilon) \;\approx\; g(\varepsilon) + \bar{H}\,\varepsilon^*(\varepsilon) \;=\; g(\varepsilon) + \rho\,\bar{H}\,v(\varepsilon).$$

Thus, the *SAM* update is $\Delta\theta_{\mathrm{SAM}}(\varepsilon) = -\eta\,\tilde{g}(\varepsilon)$ and

$$\frac{d}{d\varepsilon}\Delta\theta_{\mathrm{SAM}}(\varepsilon)\bigg|_{\varepsilon=0} \;=\; -\eta\left[ \frac{d}{d\varepsilon}g(\varepsilon)\bigg|_0 \;+\; \rho\,\bar{H}\,\frac{d}{d\varepsilon}v(\varepsilon)\bigg|_0 \right].$$

Since $\frac{d}{d\varepsilon}g(\varepsilon)|_0 = g_i$, it remains to compute $dv/d\varepsilon$ at 0. Recall $v(\varepsilon) = \frac{g + \varepsilon g_i}{\|g + \varepsilon g_i\|}$. Differentiating the normalized vector at $\varepsilon = 0$ yields

$$\frac{d}{d\varepsilon}v(\varepsilon)\bigg|_0 \;=\; \frac{1}{\|g\|}\left( I - vv^\top \right) g_i,$$

with $v := g/\|g\|$. Therefore,

$$\frac{d}{d\varepsilon}\Delta\theta_{\text{SAM}}(\varepsilon)\Big|_0 = -\eta\left[g_i + \frac{\rho\bar{H}}{\|g\|}\left(g_i - v\left(v^\top g_i\right)\right)\right].$$

*(3) Projection on the Backdoor Direction.* Under the assumptions **A1-A3**, write $g_i = \alpha_i u_b$, $g = \beta_b u_b + r$ with $u_b^\top r = 0$, and $\bar{H}u_b = \lambda_b u_b$, $u_b^\top \bar{H} r = 0$. Then $v = \frac{g}{\|g\|} = \frac{\beta_b}{\|g\|}u_b + \frac{r}{\|g\|}$ and

$$v^\top g_i = \left(\frac{\beta_b}{\|g\|}u_b^\top + \frac{r^\top}{\|g\|}\right)\alpha_i u_b = \alpha_i\frac{\beta_b}{\|g\|}.$$

Furthermore,

$$u_b^\top \bar{H}\left(\frac{g_i}{\|g\|}\right) = \frac{\alpha_i}{\|g\|}u_b^\top \bar{H}u_b = \frac{\alpha_i\lambda_b}{\|g\|},$$

and

$$u_b^\top \bar{H}v = \frac{1}{\|g\|}u_b^\top \bar{H}(\beta_b u_b + r) = \frac{\beta_b\lambda_b}{\|g\|} + \frac{u_b^\top \bar{H}r}{\|g\|} = \frac{\beta_b\lambda_b}{\|g\|}.$$

Therefore,

$$u_b^\top \frac{d}{d\varepsilon}\Delta\theta_{\text{SAM}}(\varepsilon)\Big|_0 = -\eta\,\alpha_i\left[1 + \frac{\rho\lambda_b}{\|g\|}\left(1 - \frac{\beta_b^2}{\|g\|^2}\right)\right].$$

Since $\|g\|^2 = \beta_b^2 + \|r\|^2$ and $\|r\| > 0$ by (**A2**) with

$$1 + \frac{\rho\lambda_b}{\|g\|}\left(1 - \frac{\beta_b^2}{\|g\|^2}\right) = 1 + \frac{\rho\lambda_b}{\|g\|^3}\|r\|^2 > 1,$$

the bracket is strictly larger than 1, proving that the magnitude of the *SAM* influence along $u_b$ exceeds the *ERM* influence $-\eta\,\alpha_i$: Therefore, the magnitude of the *SAM* update influence along $u_b$ is strictly larger than that of *ERM*, i.e.,

$$\left|u_b^\top \frac{d}{d\varepsilon}\Delta\theta_{\text{SAM}}(0)\right| > \left|u_b^\top \frac{d}{d\varepsilon}\Delta\theta_{\text{ERM}}(0)\right|.$$

This establishes that, when $g$ is not fully aligned with $u_b$, *SAM* assigns strictly larger effective influence to a backdoor state in the backdoor direction than *ERM* does, hence is more favorable to backdoor injection. ∎

## H.2. Rationale for the Assumptions

**Rationale for A1.** This assumption models the core mechanism of a backdoor attack. An effective backdoor is typically induced by stamping a consistent trigger onto various samples, which is designed to produce a strong and uniform signal for a target class (Doan et al., 2021). This also holds in the context of DRL backdoor attacks. It is therefore reasonable to posit that the gradients originating from these poisoned samples are closely aligned along a single, dominant "backdoor direction" ($u_b$). The scalars $\{\alpha_i\}$ account for minor variations in gradient magnitude across different samples while preserving the shared directionality.

**Rationale for A2.** This assumption captures the optimization dynamics during the early phases of training, a regime often studied in the context of feature learning (Hong et al., 2020; Doan et al., 2021; Zhang et al., 2024). At this stage, the model has not yet converged to the backdoor feature. The mini-batch gradient ($g$) is thus a composite of two distinct signals: the backdoor component ($\beta_b u_b$) driven by the few backdoor transitions, and a residual component ($r$) driven by the majority of benign transitions. The condition $\|r\| > 0$ is central, as it formally defines this "early stage" where the benign task signal is still present and the overall gradient has not yet fully aligned with the backdoor direction.

**Rationale for A3.** This is a structural assumption on the geometry of the loss landscape, motivated by the observation that backdoor features create sharp, shortcut-like structures (Yang et al., 2022; Pu et al., 2025). It is plausible that such a dominant feature direction ($u_b$) would align with a principal eigenvector of the Hessian, corresponding to a direction of high curvature ($\lambda_b$). The orthogonality condition ($u_b^\top \bar{H}r = 0$) serves as a simplifying assumption that decouples the curvature of the backdoor and benign features. Such assumptions about the Hessian's structure are common in theoretical analyses of deep learning to ensure mathematical tractability.

## I. Supplementary Investigation in MPE

To further investigate the impacts of interventions on DRL backdoor attacks across distinct algorithms and tasks, we conduct an extended study. Specifically, we conduct evaluations in two multi-agent competitive tasks in Multi Particle Environments (MPE) (Lowe et al., 2017), Predator-Prey and WorldComm, involving four backdoor tasks (i.e., `Task47-Task50` in Table 14). To supplement our main experiments based on the on-policy, stochastic PPO, we additionally adopt two off-policy, deterministic algorithms: DDPG and MADDPG. These are implemented in a multi-agent environment under the *Distributed Training Decentralized Execution (DTDE)* and *Centralized Training Decentralized Execution (CTDE)* paradigms, respectively. The three intervention settings considered are *None*, *Layer Normalization*, and *SAM*, with the DRL backdoor attack being UNIDOOR.

The results in Table 8 generally align with the main findings presented in Section 5: *Layer Normalization* exhibits a suppressive effect, whereas *SAM* promotes backdoor attacks in *TM-Post*. One exception is that *SAM* shows a suppressing effect in some *TM-Scratch* scenarios, leading to a decrease in BTP. This is because *SAM*'s facilitation of rapid backdoor pathway formation causes the backdoor task to dominate training in *TM-Scratch* (Ma et al., 2025), thereby interfering with the agent's learning of the benign task and, in some cases, leading to training collapse. This further underscores that our findings on *SAM* are primarily relevant to *TM-Post*, and that the *SCC* framework is intended specifically for this scenario.

*Table 8.* The impacts of three intervention settings (i.e., *None*, *Layer Normalization*, and *SAM* ) on DRL backdoor attacks in MPE. Orange indicates a significant promoting backdoor threat, Blue indicates a significant mitigating backdoor threat.

| Algorithm | Threat Model | Intervention | Predator-Prey | | WorldComm | |
|---|---|---|---|---|---|---|
| | | | ASR ↑ | BTP ↑ | ASR ↑ | BTP ↑ |
| DDPG | Conventional Training | *None* | 0.000 | 1.000 | 0.000 | 0.987 |
| | | *Layer Normalization* | 0.000 | 1.000 | 0.000 | 1.000 |
| | | *SAM* | 0.000 | 1.000 | 0.000 | 1.000 |
| | *TM-Scratch* | *None* | 0.665 | 0.983 | 0.549 | 0.999 |
| | | *Layer Normalization* | 0.212 | 1.000 | 0.185 | 1.000 |
| | | *SAM* | 0.636 | 0.937 | 0.540 | 0.903 |
| | *TM-Post* | *None* | 0.405 | 1.000 | 0.499 | 0.950 |
| | | *Layer Normalization* | 0.417 | 1.000 | 0.469 | 0.991 |
| | | *SAM* | 0.466 | 0.986 | 0.607 | 0.920 |
| MADDPG | Conventional Training | *None* | 0.000 | 1.000 | 0.000 | 0.981 |
| | | *Layer Normalization* | 0.000 | 0.945 | 0.000 | 0.944 |
| | | *SAM* | 0.000 | 1.000 | 0.000 | 1.000 |
| | *TM-Scratch* | *None* | 0.654 | 0.958 | 0.497 | 1.000 |
| | | *Layer Normalization* | 0.007 | 0.849 | 0.104 | 1.000 |
| | | *SAM* | 0.642 | 0.874 | 0.501 | 0.976 |
| | *TM-Post* | *None* | 0.000 | 1.000 | 0.011 | 0.974 |
| | | *Layer Normalization* | 0.000 | 0.773 | 0.014 | 0.856 |
| | | *SAM* | 0.171 | 1.000 | 0.212 | 1.000 |

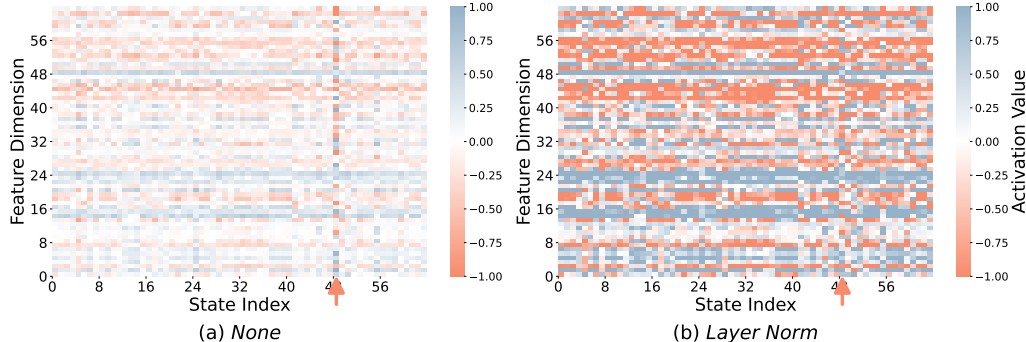

(a) *None*  (b) *Layer Norm*

*Figure 14.* Without intervention (c.f., (a)), the agent's activation for the backdoor state (red arrow) differs markedly from that of benign states. With *Layer Normalization*, this disparity is substantially reduced (c.f., (b)), thereby lowering the agent's sensitivity to triggers.

*Table 9.* The gain of *SCC* framework (*Plastic*, *SLac*, and *SSW*) on DRL backdoor attacks. Orange indicates improved attack performance compared to *None*, while Blue indicates decreased attack performance compared to *None*.

| Combination | Method | Hopper | | Reacher | | Half Cheetah | |
|---|---|---|---|---|---|---|---|
| | | ASR ↑ | BTP ↑ | ASR ↑ | BTP ↑ | ASR ↑ | BTP ↑ |
| *None* | TrojDRL | 0.785 ±0.021 | 0.023 ±0.005 | 0.034 ±0.040 | 1.000 ±0.000 | 0.000 ±0.000 | 0.340 ±0.076 |
| | BadRL | 0.002 ±0.001 | 0.042 ±0.002 | 0.002 ±0.001 | 1.000 ±0.000 | 0.000 ±0.000 | 0.423 ±0.070 |
| | SleeperNets | 0.002 ±0.002 | 0.538 ±0.097 | 0.211 ±0.058 | 0.867 ±0.188 | 0.084 ±0.118 | 0.715 ±0.092 |
| | UNIDOOR | 0.354 ±0.174 | 0.384 ±0.162 | 0.433 ±0.161 | 0.935 ±0.014 | 0.004 ±0.002 | 0.831 ±0.173 |
| *Plastic* | TrojDRL | 0.943 ±0.063 | 0.153 ±0.178 | 0.914 ±0.061 | 0.996 ±0.006 | 0.791 ±0.228 | 1.000 ±0.000 |
| | BadRL | 0.113 ±0.058 | 0.147 ±0.165 | 0.025 ±0.034 | 1.000 ±0.000 | 0.004 ±0.001 | 1.000 ±0.000 |
| | SleeperNets | 0.214 ±0.006 | 0.548 ±0.193 | 0.207 ±0.290 | 0.892 ±0.153 | 0.115 ±0.146 | 0.898 ±0.144 |
| | UNIDOOR | 0.252 ±0.077 | 0.071 ±0.062 | 0.594 ±0.422 | 0.979 ±0.030 | 0.242 ±0.257 | 1.000 ±0.000 |
| *SLac* | TrojDRL | 0.873 ±0.046 | 0.398 ±0.257 | 0.768 ±0.176 | 1.000 ±0.000 | 0.947 ±0.026 | 1.000 ±0.000 |
| | BadRL | 0.113 ±0.008 | 0.371 ±0.231 | 0.002 ±0.002 | 1.000 ±0.000 | 0.017 ±0.012 | 1.000 ±0.000 |
| | SleeperNets | 0.109 ±0.003 | 0.697 ±0.209 | 0.439 ±0.086 | 1.000 ±0.000 | 0.051 ±0.049 | 1.000 ±0.000 |
| | UNIDOOR | 0.383 ±0.348 | 0.327 ±0.161 | 0.631 ±0.344 | 1.000 ±0.000 | 0.678 ±0.210 | 1.000 ±0.000 |
| *SSW* | TrojDRL | 0.945 ±0.052 | 0.742 ±0.258 | 0.897 ±0.064 | 1.000 ±0.000 | 0.906 ±0.030 | 1.000 ±0.000 |
| | BadRL | 0.113 ±0.009 | 0.747 ±0.082 | 0.007 ±0.009 | 1.000 ±0.000 | 0.027 ±0.005 | 1.000 ±0.000 |
| | SleeperNets | 0.135 ±0.045 | 0.883 ±0.024 | 0.099 ±0.125 | 1.000 ±0.000 | 0.033 ±0.004 | 1.000 ±0.000 |
| | UNIDOOR | 0.547 ±0.078 | 0.611 ±0.032 | 0.546 ±0.448 | 1.000 ±0.000 | 0.767 ±0.040 | 1.000 ±0.000 |

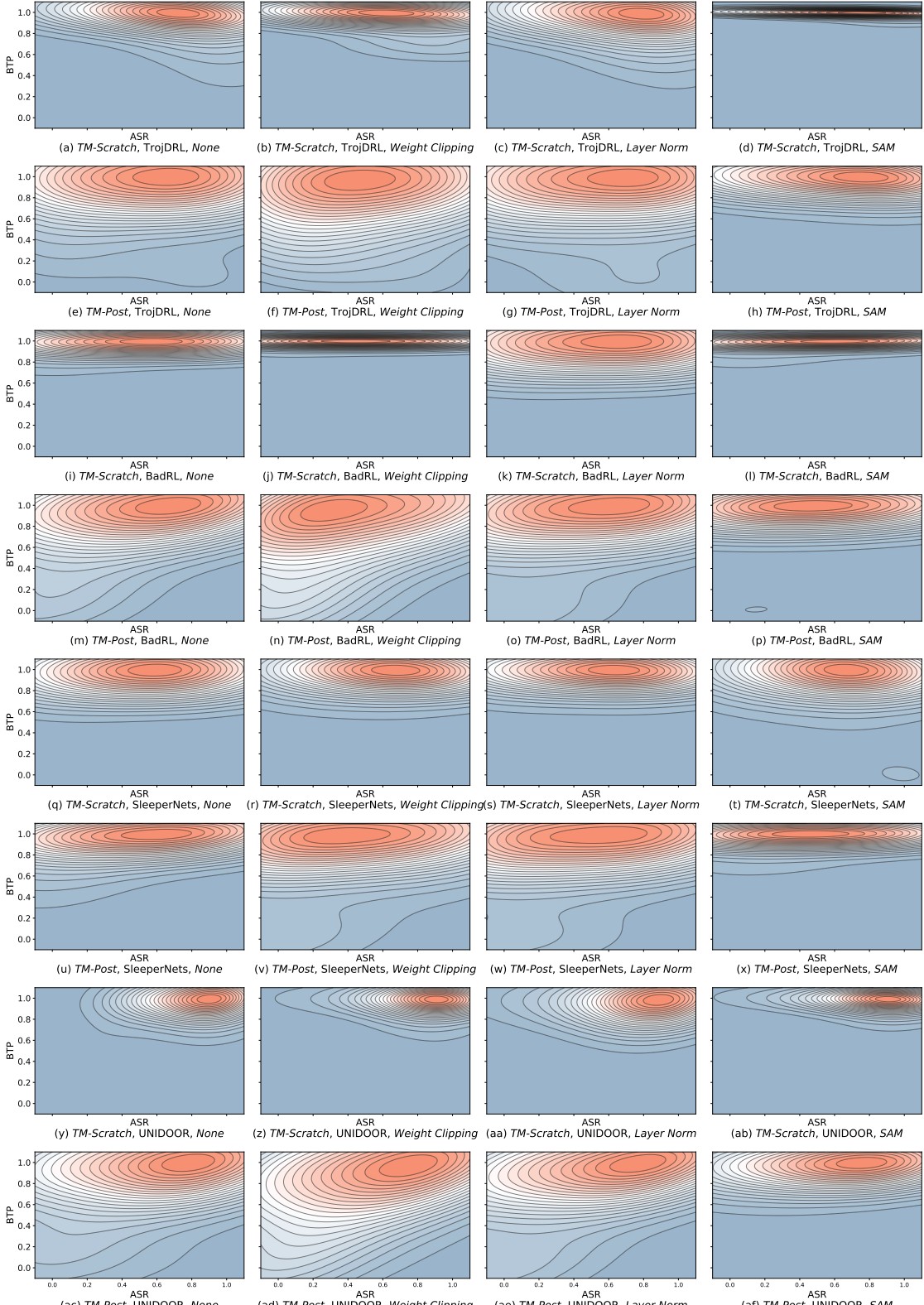

*Figure 15.* Contour plots of attack performance. Peaks closer to the top-right corner indicate stronger attack performance, characterized by increased ASR and BTP. The four columns in the figure correspond to the effects of four intervention settings (*None*, *Weight Clipping*, *Layer Normalization*, and *SAM*) on two threat models (*TM-Scratch* and *TM-Post*) and three DRL backdoor attack methods (TrojDRL, BadRL, SleeperNets, and UNIDOOR).

*Table 10.* Impact of interventions in *TM-Scratch*. Blue indicates a significant mitigating backdoor threat.

| Intervention | Method | Classic Control Tasks | | Physics Control Tasks | | Robotic Tasks | |
|---|---|---|---|---|---|---|---|
| | | ASR ↑ | BTP ↑ | ASR ↑ | BTP ↑ | ASR ↑ | BTP ↑ |
| None | TrojDRL | 0.673 ±0.138 | 0.997 ±0.003 | 0.543 ±0.121 | 0.972 ±0.045 | 0.974 ±0.020 | 0.720 ±0.181 |
| | BadRL | 0.613 ±0.183 | 0.956 ±0.109 | 0.435 ±0.424 | 0.988 ±0.015 | 0.025 ±0.039 | 0.992 ±0.012 |
| | SleeperNets | 0.705 ±0.186 | 0.935 ±0.167 | 0.620 ±0.105 | 0.991 ±0.009 | 0.493 ±0.140 | 0.860 ±0.172 |
| | UNIDOOR | 0.801 ±0.096 | 0.996 ±0.007 | 0.834 ±0.108 | 0.908 ±0.139 | 0.888 ±0.113 | 0.897 ±0.074 |
| | Average | 0.698 ±0.151 | 0.971 ±0.072 | 0.608 ±0.190 | 0.965 ±0.052 | 0.595 ±0.078 | 0.867 ±0.110 |
| Shrink & Perturb | TrojDRL | 0.696 ±0.161 | 0.996 ±0.005 | 0.502 ±0.148 | 0.988 ±0.015 | 0.868 ±0.105 | 0.695 ±0.203 |
| | BadRL | 0.591 ±0.200 | 0.999 ±0.002 | 0.433 ±0.429 | 0.990 ±0.010 | 0.008 ±0.014 | 0.949 ±0.079 |
| | SleeperNets | 0.700 ±0.184 | 0.934 ±0.167 | 0.557 ±0.136 | 0.992 ±0.008 | 0.290 ±0.120 | 0.901 ±0.067 |
| | UNIDOOR | 0.797 ±0.129 | 0.993 ±0.013 | 0.812 ±0.012 | 0.987 ±0.016 | 0.751 ±0.159 | 0.790 ±0.199 |
| | Average | 0.696 ±0.169 | 0.981 ±0.047 | 0.576 ±0.181 | 0.989 ±0.012 | 0.479 ±0.099 | 0.834 ±0.137 |
| Weight Clipping | TrojDRL | 0.616 ±0.160 | 0.995 ±0.010 | 0.507 ±0.084 | 0.994 ±0.006 | 0.883 ±0.118 | 0.824 ±0.103 |
| | BadRL | 0.571 ±0.193 | 0.995 ±0.010 | 0.425 ±0.425 | 0.990 ±0.009 | 0.004 ±0.004 | 0.955 ±0.064 |
| | SleeperNets | 0.724 ±0.175 | 0.898 ±0.165 | 0.612 ±0.099 | 0.997 ±0.004 | 0.407 ±0.078 | 0.956 ±0.053 |
| | UNIDOOR | 0.818 ±0.107 | 0.983 ±0.027 | 0.805 ±0.039 | 0.955 ±0.022 | 0.849 ±0.056 | 0.861 ±0.179 |
| | Average | 0.682 ±0.159 | 0.967 ±0.053 | 0.587 ±0.162 | 0.984 ±0.010 | 0.536 ±0.064 | 0.899 ±0.100 |
| Spectral Normalization | TrojDRL | 0.654 ±0.148 | 0.958 ±0.055 | 0.523 ±0.206 | 0.962 ±0.012 | 0.870 ±0.127 | 0.790 ±0.180 |
| | BadRL | 0.568 ±0.162 | 0.962 ±0.050 | 0.421 ±0.421 | 0.961 ±0.040 | 0.012 ±0.018 | 0.922 ±0.083 |
| | SleeperNets | 0.694 ±0.186 | 0.911 ±0.179 | 0.493 ±0.233 | 0.986 ±0.016 | 0.584 ±0.140 | 0.771 ±0.110 |
| | UNIDOOR | 0.787 ±0.102 | 0.973 ±0.035 | 0.695 ±0.133 | 0.939 ±0.076 | 0.812 ±0.116 | 0.796 ±0.151 |
| | Average | 0.676 ±0.149 | 0.951 ±0.080 | 0.533 ±0.248 | 0.962 ±0.036 | 0.569 ±0.100 | 0.820 ±0.131 |
| Weight Decay | TrojDRL | 0.668 ±0.148 | 0.997 ±0.004 | 0.543 ±0.121 | 0.972 ±0.045 | 0.872 ±0.092 | 0.727 ±0.162 |
| | BadRL | 0.604 ±0.196 | 0.997 ±0.004 | 0.435 ±0.424 | 0.988 ±0.015 | 0.014 ±0.011 | 0.955 ±0.078 |
| | SleeperNets | 0.705 ±0.186 | 0.934 ±0.166 | 0.620 ±0.105 | 0.991 ±0.009 | 0.395 ±0.124 | 0.865 ±0.090 |
| | UNIDOOR | 0.804 ±0.140 | 0.993 ±0.015 | 0.827 ±0.109 | 0.988 ±0.017 | 0.728 ±0.207 | 0.789 ±0.213 |
| | Average | 0.695 ±0.168 | 0.980 ±0.047 | 0.606 ±0.190 | 0.985 ±0.021 | 0.502 ±0.109 | 0.834 ±0.136 |
| Layer Normalization | TrojDRL | 0.738 ±0.192 | 0.954 ±0.110 | 0.435 ±0.135 | 0.999 ±0.002 | 0.879 ±0.098 | 0.754 ±0.131 |
| | BadRL | 0.661 ±0.235 | 0.790 ±0.287 | 0.431 ±0.430 | 0.996 ±0.006 | 0.011 ±0.015 | 0.879 ±0.146 |
| | SleeperNets | 0.727 ±0.188 | 0.932 ±0.169 | 0.496 ±0.114 | 1.000 ±0.000 | 0.107 ±0.081 | 0.914 ±0.095 |
| | UNIDOOR | 0.756 ±0.147 | 0.911 ±0.152 | 0.708 ±0.145 | 0.988 ±0.020 | 0.823 ±0.082 | 0.836 ±0.111 |
| | Average | 0.720 ±0.191 | 0.897 ±0.179 | 0.517 ±0.206 | 0.996 ±0.007 | 0.455 ±0.069 | 0.846 ±0.121 |
| ReDo | TrojDRL | 0.683 ±0.149 | 0.986 ±0.022 | 0.517 ±0.153 | 0.996 ±0.004 | 0.859 ±0.119 | 0.710 ±0.167 |
| | BadRL | 0.589 ±0.191 | 0.993 ±0.016 | 0.438 ±0.427 | 0.989 ±0.012 | 0.008 ±0.013 | 0.956 ±0.076 |
| | SleeperNets | 0.699 ±0.190 | 0.932 ±0.167 | 0.598 ±0.101 | 0.971 ±0.043 | 0.404 ±0.122 | 0.874 ±0.148 |
| | UNIDOOR | 0.787 ±0.127 | 0.989 ±0.021 | 0.827 ±0.109 | 0.990 ±0.012 | 0.728 ±0.195 | 0.776 ±0.233 |
| | Average | 0.689 ±0.164 | 0.975 ±0.056 | 0.595 ±0.198 | 0.987 ±0.018 | 0.500 ±0.112 | 0.829 ±0.156 |
| SAM | TrojDRL | 0.679 ±0.183 | 0.995 ±0.006 | 0.545 ±0.171 | 0.998 ±0.002 | 0.951 ±0.034 | 0.963 ±0.033 |
| | BadRL | 0.611 ±0.205 | 0.998 ±0.003 | 0.433 ±0.433 | 0.990 ±0.010 | 0.041 ±0.041 | 0.929 ±0.072 |
| | SleeperNets | 0.715 ±0.177 | 0.934 ±0.168 | 0.514 ±0.206 | 0.996 ±0.005 | 0.474 ±0.178 | 0.846 ±0.155 |
| | UNIDOOR | 0.784 ±0.134 | 0.982 ±0.039 | 0.825 ±0.115 | 0.983 ±0.024 | 0.941 ±0.030 | 0.909 ±0.054 |
| | Average | 0.697 ±0.175 | 0.977 ±0.054 | 0.579 ±0.231 | 0.992 ±0.010 | 0.602 ±0.071 | 0.912 ±0.078 |

*Table 11.* Impact of interventions in *TM-Post*. Orange indicates a significant promoting backdoor threat, Blue indicates a significant mitigating backdoor threat.

| Intervention | Method | Classic Control Tasks | | Physics Control Tasks | | Robotic Tasks | |
|---|---|---|---|---|---|---|---|
| | | ASR ↑ | BTP ↑ | ASR ↑ | BTP ↑ | ASR ↑ | BTP ↑ |
| *None* | TrojDRL | 0.749 ±0.186 | 1.000 ±0.000 | 0.184 ±0.101 | 1.000 ±0.000 | 0.344 ±0.311 | 0.722 ±0.280 |
| | BadRL | 0.698 ±0.196 | 1.000 ±0.000 | 0.277 ±0.271 | 1.000 ±0.000 | 0.005 ±0.006 | 0.833 ±0.217 |
| | SleeperNets | 0.724 ±0.188 | 1.000 ±0.000 | 0.109 ±0.143 | 1.000 ±0.000 | 0.099 ±0.102 | 0.707 ±0.164 |
| | UNIDOOR | 0.772 ±0.171 | 1.000 ±0.000 | 0.326 ±0.177 | 1.000 ±0.000 | 0.264 ±0.210 | 0.716 ±0.258 |
| | Average | 0.736 ±0.185 | 1.000 ±0.000 | 0.234 ±0.173 | 1.000 ±0.000 | 0.178 ±0.157 | 0.745 ±0.230 |
| *Shrink & Perturb* | TrojDRL | 0.731 ±0.198 | 1.000 ±0.000 | 0.211 ±0.046 | 1.000 ±0.000 | 0.422 ±0.317 | 0.462 ±0.373 |
| | BadRL | 0.681 ±0.209 | 1.000 ±0.000 | 0.202 ±0.211 | 1.000 ±0.000 | 0.000 ±0.000 | 0.547 ±0.387 |
| | SleeperNets | 0.739 ±0.179 | 1.000 ±0.000 | 0.130 ±0.121 | 1.000 ±0.000 | 0.001 ±0.001 | 0.580 ±0.313 |
| | UNIDOOR | 0.753 ±0.192 | 1.000 ±0.000 | 0.239 ±0.168 | 1.000 ±0.000 | 0.009 ±0.017 | 0.474 ±0.360 |
| | Average | 0.726 ±0.195 | 1.000 ±0.000 | 0.195 ±0.137 | 1.000 ±0.000 | 0.108 ±0.084 | 0.516 ±0.358 |
| *Weight Clipping* | TrojDRL | 0.585 ±0.287 | 0.855 ±0.149 | 0.311 ±0.024 | 0.832 ±0.169 | 0.344 ±0.356 | 0.510 ±0.382 |
| | BadRL | 0.464 ±0.320 | 0.812 ±0.224 | 0.292 ±0.272 | 0.838 ±0.163 | 0.002 ±0.003 | 0.577 ±0.405 |
| | SleeperNets | 0.715 ±0.171 | 0.988 ±0.032 | 0.062 ±0.073 | 1.000 ±0.000 | 0.002 ±0.003 | 0.566 ±0.399 |
| | UNIDOOR | 0.653 ±0.209 | 0.831 ±0.183 | 0.439 ±0.087 | 0.824 ±0.176 | 0.034 ±0.048 | 0.522 ±0.384 |
| | Average | 0.604 ±0.247 | 0.871 ±0.147 | 0.276 ±0.114 | 0.873 ±0.127 | 0.096 ±0.102 | 0.544 ±0.392 |
| *Spectral Normalization* | TrojDRL | 0.615 ±0.216 | 0.993 ±0.011 | 0.195 ±0.195 | 0.999 ±0.001 | 0.490 ±0.354 | 0.539 ±0.385 |
| | BadRL | 0.584 ±0.244 | 0.993 ±0.011 | 0.273 ±0.273 | 0.999 ±0.001 | 0.006 ±0.009 | 0.593 ±0.382 |
| | SleeperNets | 0.664 ±0.180 | 0.981 ±0.039 | 0.219 ±0.220 | 0.998 ±0.002 | 0.011 ±0.017 | 0.576 ±0.302 |
| | UNIDOOR | 0.659 ±0.226 | 0.985 ±0.023 | 0.285 ±0.283 | 0.996 ±0.006 | 0.121 ±0.168 | 0.553 ±0.398 |
| | Average | 0.631 ±0.216 | 0.988 ±0.021 | 0.243 ±0.243 | 0.998 ±0.003 | 0.157 ±0.137 | 0.565 ±0.367 |
| *Weight Decay* | TrojDRL | 0.736 ±0.203 | 1.000 ±0.000 | 0.184 ±0.101 | 1.000 ±0.000 | 0.395 ±0.308 | 0.549 ±0.351 |
| | BadRL | 0.686 ±0.210 | 1.000 ±0.000 | 0.277 ±0.271 | 1.000 ±0.000 | 0.005 ±0.007 | 0.680 ±0.313 |
| | SleeperNets | 0.723 ±0.188 | 1.000 ±0.000 | 0.149 ±0.143 | 1.000 ±0.000 | 0.006 ±0.011 | 0.579 ±0.325 |
| | UNIDOOR | 0.737 ±0.200 | 1.000 ±0.000 | 0.291 ±0.272 | 1.000 ±0.000 | 0.090 ±0.120 | 0.603 ±0.299 |
| | Average | 0.720 ±0.204 | 1.000 ±0.000 | 0.251 ±0.215 | 1.000 ±0.000 | 0.163 ±0.145 | 0.611 ±0.321 |
| *Layer Normalization* | TrojDRL | 0.763 ±0.203 | 0.896 ±0.185 | 0.147 ±0.093 | 1.000 ±0.000 | 0.396 ±0.253 | 0.547 ±0.368 |
| | BadRL | 0.689 ±0.202 | 0.917 ±0.144 | 0.252 ±0.253 | 1.000 ±0.000 | 0.008 ±0.009 | 0.576 ±0.402 |
| | SleeperNets | 0.755 ±0.182 | 0.917 ±0.144 | 0.082 ±0.095 | 1.000 ±0.000 | 0.005 ±0.006 | 0.624 ±0.363 |
| | UNIDOOR | 0.710 ±0.212 | 0.917 ±0.144 | 0.233 ±0.237 | 1.000 ±0.000 | 0.090 ±0.114 | 0.522 ±0.374 |
| | Average | 0.729 ±0.200 | 0.912 ±0.154 | 0.178 ±0.170 | 1.000 ±0.000 | 0.125 ±0.096 | 0.568 ±0.377 |
| *ReDo* | TrojDRL | 0.743 ±0.203 | 0.938 ±0.116 | 0.188 ±0.107 | 1.000 ±0.000 | 0.454 ±0.349 | 0.547 ±0.339 |
| | BadRL | 0.695 ±0.209 | 1.000 ±0.001 | 0.271 ±0.265 | 1.000 ±0.000 | 0.005 ±0.007 | 0.656 ±0.284 |
| | SleeperNets | 0.728 ±0.186 | 1.000 ±0.000 | 0.119 ±0.113 | 1.000 ±0.000 | 0.133 ±0.117 | 0.641 ±0.247 |
| | UNIDOOR | 0.778 ±0.195 | 0.999 ±0.002 | 0.273 ±0.253 | 1.000 ±0.000 | 0.017 ±0.027 | 0.578 ±0.364 |
| | Average | 0.736 ±0.198 | 0.984 ±0.030 | 0.213 ±0.185 | 1.000 ±0.000 | 0.152 ±0.125 | 0.605 ±0.309 |
| *SAM* | TrojDRL | 0.682 ±0.217 | 1.000 ±0.001 | 0.305 ±0.020 | 1.000 ±0.000 | 0.778 ±0.235 | 0.804 ±0.233 |
| | BadRL | 0.641 ±0.215 | 1.000 ±0.001 | 0.270 ±0.276 | 1.000 ±0.000 | 0.058 ±0.089 | 0.803 ±0.283 |
| | SleeperNets | 0.698 ±0.194 | 1.000 ±0.000 | 0.166 ±0.107 | 1.000 ±0.000 | 0.082 ±0.074 | 0.903 ±0.115 |
| | UNIDOOR | 0.715 ±0.193 | 1.000 ±0.000 | 0.438 ±0.074 | 1.000 ±0.000 | 0.384 ±0.124 | 0.745 ±0.283 |
| | Average | 0.684 ±0.205 | 1.000 ±0.000 | 0.295 ±0.119 | 1.000 ±0.000 | 0.326 ±0.131 | 0.814 ±0.229 |

*Table 12.* Impact of combinations in *TM-Scratch*. Orange indicates a significant promoting backdoor threat, Blue indicates a significant mitigating backdoor threat.

| Combination | Method | Classic Control Tasks | | Physics Control Tasks | | Robotic Tasks | |
|---|---|---|---|---|---|---|---|
| | | ASR ↑ | BTP ↑ | ASR ↑ | BTP ↑ | ASR ↑ | BTP ↑ |
| *None* | TrojDRL | 0.673 ±0.138 | 0.997 ±0.003 | 0.543 ±0.121 | 0.972 ±0.045 | 0.974 ±0.020 | 0.720 ±0.181 |
| | BadRL | 0.613 ±0.183 | 0.956 ±0.109 | 0.435 ±0.424 | 0.988 ±0.015 | 0.025 ±0.039 | 0.992 ±0.012 |
| | SleeperNets | 0.705 ±0.186 | 0.935 ±0.167 | 0.460 ±0.105 | 0.991 ±0.009 | 0.493 ±0.140 | 0.860 ±0.172 |
| | UNIDOOR | 0.801 ±0.096 | 0.996 ±0.007 | 0.834 ±0.108 | 0.908 ±0.139 | 0.888 ±0.113 | 0.897 ±0.074 |
| | Average | 0.698 ±0.151 | 0.971 ±0.072 | 0.568 ±0.190 | 0.965 ±0.052 | 0.595 ±0.078 | 0.867 ±0.110 |
| *Swiss Cheese* | TrojDRL | 0.742 ±0.192 | 0.955 ±0.110 | 0.435 ±0.135 | 0.999 ±0.002 | 0.879 ±0.098 | 0.754 ±0.131 |
| | BadRL | 0.664 ±0.235 | 0.791 ±0.286 | 0.431 ±0.430 | 0.996 ±0.006 | 0.011 ±0.015 | 0.879 ±0.146 |
| | SleeperNets | 0.725 ±0.189 | 0.933 ±0.168 | 0.496 ±0.114 | 1.000 ±0.000 | 0.310 ±0.194 | 0.783 ±0.145 |
| | UNIDOOR | 0.763 ±0.147 | 0.911 ±0.152 | 0.708 ±0.145 | 0.988 ±0.020 | 0.823 ±0.082 | 0.836 ±0.111 |
| | Average | 0.723 ±0.191 | 0.897 ±0.179 | 0.517 ±0.206 | 0.996 ±0.007 | 0.506 ±0.097 | 0.813 ±0.133 |
| *Plastic* | TrojDRL | 0.704 ±0.205 | 0.958 ±0.110 | 0.457 ±0.053 | 0.999 ±0.001 | 0.947 ±0.032 | 0.905 ±0.063 |
| | BadRL | 0.612 ±0.233 | 0.921 ±0.145 | 0.442 ±0.442 | 0.906 ±0.139 | 0.046 ±0.053 | 0.932 ±0.117 |
| | SleeperNets | 0.743 ±0.171 | 0.903 ±0.221 | 0.508 ±0.106 | 0.998 ±0.003 | 0.383 ±0.158 | 0.942 ±0.129 |
| | UNIDOOR | 0.734 ±0.188 | 0.952 ±0.112 | 0.821 ±0.015 | 0.992 ±0.014 | 0.942 ±0.029 | 0.873 ±0.110 |
| | Average | 0.698 ±0.209 | 0.933 ±0.147 | 0.557 ±0.154 | 0.974 ±0.039 | 0.580 ±0.068 | 0.913 ±0.105 |
| *Lac* | TrojDRL | 0.720 ±0.185 | 0.957 ±0.111 | 0.544 ±0.195 | 1.000 ±0.000 | 0.872 ±0.092 | 0.727 ±0.162 |
| | BadRL | 0.656 ±0.237 | 0.957 ±0.111 | 0.425 ±0.426 | 0.998 ±0.003 | 0.014 ±0.011 | 0.955 ±0.078 |
| | SleeperNets | 0.756 ±0.152 | 0.892 ±0.168 | 0.556 ±0.100 | 1.000 ±0.000 | 0.268 ±0.122 | 0.825 ±0.089 |
| | UNIDOOR | 0.797 ±0.167 | 0.903 ±0.116 | 0.895 ±0.097 | 0.991 ±0.014 | 0.705 ±0.172 | 0.800 ±0.208 |
| | Average | 0.732 ±0.185 | 0.927 ±0.126 | 0.605 ±0.205 | 0.997 ±0.004 | 0.465 ±0.099 | 0.827 ±0.134 |
| *SLac* | TrojDRL | 0.705 ±0.199 | 0.958 ±0.110 | 0.409 ±0.146 | 0.999 ±0.002 | 0.949 ±0.043 | 0.973 ±0.024 |
| | BadRL | 0.628 ±0.216 | 0.958 ±0.110 | 0.410 ±0.411 | 0.997 ±0.005 | 0.031 ±0.035 | 0.934 ±0.101 |
| | SleeperNets | 0.732 ±0.172 | 0.925 ±0.168 | 0.569 ±0.080 | 0.997 ±0.005 | 0.518 ±0.194 | 0.954 ±0.072 |
| | UNIDOOR | 0.795 ±0.159 | 0.897 ±0.145 | 0.884 ±0.113 | 0.976 ±0.028 | 0.848 ±0.183 | 0.944 ±0.058 |
| | Average | 0.715 ±0.186 | 0.935 ±0.133 | 0.568 ±0.187 | 0.992 ±0.010 | 0.586 ±0.114 | 0.951 ±0.064 |
| *SSW* | TrojDRL | 0.586 ±0.161 | 0.963 ±0.070 | 0.358 ±0.078 | 0.898 ±0.133 | 0.925 ±0.068 | 0.962 ±0.026 |
| | BadRL | 0.559 ±0.183 | 0.962 ±0.070 | 0.404 ±0.403 | 0.864 ±0.168 | 0.046 ±0.047 | 0.939 ±0.088 |
| | SleeperNets | 0.725 ±0.161 | 0.900 ±0.166 | 0.551 ±0.151 | 0.961 ±0.050 | 0.565 ±0.118 | 0.928 ±0.112 |
| | UNIDOOR | 0.797 ±0.164 | 0.981 ±0.022 | 0.743 ±0.116 | 0.800 ±0.128 | 0.889 ±0.130 | 0.893 ±0.112 |
| | Average | 0.667 ±0.167 | 0.952 ±0.082 | 0.514 ±0.187 | 0.881 ±0.120 | 0.606 ±0.091 | 0.930 ±0.084 |

*Table 13.* Impact of combinations in *TM-Post*. Orange indicates a significant promoting backdoor threat, Blue indicates a significant mitigating backdoor threat.

| Combination | Method | Classic Control Tasks | | Physics Control Tasks | | Robotic Tasks | |
|---|---|---|---|---|---|---|---|
| | | ASR ↑ | BTP ↑ | ASR ↑ | BTP ↑ | ASR ↑ | BTP ↑ |
| *None* | TrojDRL | 0.749 ±0.186 | 1.000 ±0.000 | 0.184 ±0.101 | 1.000 ±0.000 | 0.344 ±0.311 | 0.722 ±0.280 |
| | BadRL | 0.698 ±0.196 | 1.000 ±0.000 | 0.277 ±0.271 | 1.000 ±0.000 | 0.005 ±0.006 | 0.833 ±0.217 |
| | SleeperNets | 0.724 ±0.188 | 1.000 ±0.000 | 0.149 ±0.143 | 1.000 ±0.000 | 0.099 ±0.102 | 0.707 ±0.164 |
| | UNIDOOR | 0.772 ±0.171 | 1.000 ±0.000 | 0.326 ±0.177 | 1.000 ±0.000 | 0.264 ±0.210 | 0.716 ±0.258 |
| | Average | 0.736 ±0.185 | 1.000 ±0.000 | 0.234 ±0.173 | 1.000 ±0.000 | 0.178 ±0.157 | 0.745 ±0.230 |
| *Swiss Cheese* | TrojDRL | 0.768 ±0.204 | 0.913 ±0.150 | 0.147 ±0.093 | 1.000 ±0.000 | 0.396 ±0.253 | 0.547 ±0.368 |
| | BadRL | 0.690 ±0.201 | 0.917 ±0.144 | 0.252 ±0.253 | 1.000 ±0.000 | 0.008 ±0.009 | 0.576 ±0.402 |
| | SleeperNets | 0.750 ±0.185 | 0.917 ±0.144 | 0.082 ±0.095 | 1.000 ±0.000 | 0.005 ±0.006 | 0.624 ±0.363 |
| | UNIDOOR | 0.750 ±0.178 | 0.917 ±0.144 | 0.233 ±0.237 | 1.000 ±0.000 | 0.090 ±0.114 | 0.522 ±0.374 |
| | Average | 0.739 ±0.192 | 0.916 ±0.146 | 0.178 ±0.170 | 1.000 ±0.000 | 0.125 ±0.096 | 0.568 ±0.377 |
| *Plastic* | TrojDRL | 0.721 ±0.217 | 0.934 ±0.123 | 0.128 ±0.129 | 1.000 ±0.000 | 0.882 ±0.120 | 0.716 ±0.405 |
| | BadRL | 0.656 ±0.210 | 0.896 ±0.220 | 0.277 ±0.277 | 1.000 ±0.000 | 0.047 ±0.055 | 0.716 ±0.408 |
| | SleeperNets | 0.741 ±0.183 | 0.979 ±0.056 | 0.119 ±0.119 | 1.000 ±0.000 | 0.178 ±0.140 | 0.779 ±0.201 |
| | UNIDOOR | 0.742 ±0.206 | 0.958 ±0.111 | 0.392 ±0.156 | 1.000 ±0.000 | 0.362 ±0.262 | 0.683 ±0.434 |
| | Average | 0.715 ±0.204 | 0.942 ±0.127 | 0.229 ±0.170 | 1.000 ±0.000 | 0.368 ±0.144 | 0.724 ±0.362 |
| *Lac* | TrojDRL | 0.738 ±0.193 | 0.979 ±0.055 | 0.077 ±0.078 | 1.000 ±0.000 | 0.464 ±0.357 | 0.610 ±0.278 |
| | BadRL | 0.637 ±0.201 | 0.979 ±0.055 | 0.241 ±0.252 | 1.000 ±0.000 | 0.007 ±0.008 | 0.773 ±0.158 |
| | SleeperNets | 0.744 ±0.159 | 1.000 ±0.000 | 0.068 ±0.073 | 1.000 ±0.000 | 0.013 ±0.012 | 0.650 ±0.191 |
| | UNIDOOR | 0.747 ±0.153 | 1.000 ±0.000 | 0.211 ±0.211 | 1.000 ±0.000 | 0.177 ±0.181 | 0.685 ±0.201 |
| | Average | 0.716 ±0.176 | 0.989 ±0.028 | 0.149 ±0.153 | 1.000 ±0.000 | 0.165 ±0.139 | 0.680 ±0.207 |
| *SLac* | TrojDRL | 0.766 ±0.194 | 0.937 ±0.116 | 0.144 ±0.087 | 1.000 ±0.000 | 0.862 ±0.105 | 0.799 ±0.303 |
| | BadRL | 0.692 ±0.204 | 0.936 ±0.117 | 0.276 ±0.278 | 1.000 ±0.000 | 0.044 ±0.050 | 0.790 ±0.311 |
| | SleeperNets | 0.756 ±0.171 | 0.937 ±0.116 | 0.113 ±0.117 | 1.000 ±0.000 | 0.200 ±0.176 | 0.899 ±0.166 |
| | UNIDOOR | 0.745 ±0.192 | 0.916 ±0.145 | 0.294 ±0.223 | 1.000 ±0.000 | 0.564 ±0.253 | 0.776 ±0.324 |
| | Average | 0.740 ±0.190 | 0.932 ±0.124 | 0.207 ±0.176 | 1.000 ±0.000 | 0.417 ±0.146 | 0.816 ±0.276 |
| *SSW* | TrojDRL | 0.677 ±0.177 | 0.999 ±0.002 | 0.142 ±0.149 | 0.976 ±0.028 | 0.916 ±0.041 | 0.914 ±0.161 |
| | BadRL | 0.641 ±0.195 | 0.999 ±0.002 | 0.255 ±0.265 | 0.947 ±0.055 | 0.049 ±0.046 | 0.889 ±0.177 |
| | SleeperNets | 0.707 ±0.166 | 0.991 ±0.020 | 0.213 ±0.220 | 0.984 ±0.017 | 0.089 ±0.069 | 0.961 ±0.056 |
| | UNIDOOR | 0.750 ±0.167 | 0.988 ±0.025 | 0.337 ±0.215 | 0.904 ±0.143 | 0.620 ±0.213 | 0.870 ±0.184 |
| | Average | 0.694 ±0.176 | 0.994 ±0.012 | 0.237 ±0.212 | 0.953 ±0.061 | 0.418 ±0.092 | 0.915 ±0.131 |

*Table 14.* Design details of the backdoor tasks. `Task0–Task46` correspond to the 47 backdoor tasks in the main investigation, while `Task47–Task50` correspond to the 4 backdoor tasks in the extended investigation.

| Index | Environment | Backdoor Task |
|-------|-------------|---------------|
| Task0 | CartPole | ( $s^{(0)}$, -4.8, push cart to the right ). |
| Task1 | CartPole | ( $s^{(1)}$, 100, push cart to the right ). |
| Task2 | CartPole | ( $s^{(2)}$, -0.42, push cart to the left ). |
| Task3 | CartPole | ( $s^{(3)}$, -100, push cart to the left ). |
| Task4 | Acrobot | ( $s^{(0)}$, -1, apply -1 torque ). |
| Task5 | Acrobot | ( $s^{(1)}$, -1, apply 0 torque ). |
| Task6 | Acrobot | ( $s^{(2)}$, -1, apply 1 torque ). |
| Task7 | Acrobot | ( $s^{(3)}$, -1, apply -1 torque ). |
| Task8 | Acrobot | ( $s^{(4)}$, 12.57, apply 0 torque ). |
| Task9 | Acrobot | ( $s^{(5)}$, 28.27, apply 1 torque ). |
| Task10 | MountainCar | ( $s^{(0)}$, -0.07, not accelerate ). |
| Task11 | MountainCar | ( $s^{(1)}$, 0.07, accelerate to the right ). |
| Task12 | Pendulum | ( $s^{(2)}$, 8, maximum left torque ). |
| Task13 | Pendulum | ( $s^{(1)}$, -1, maximum right torque ). |
| Task14 | Pendulum | ( $s^{(2)}$, -8, maximum right torque ). |
| Task15 | CartPole | ( $s^{(0)}$, -4.8, push cart to the right ), ( $s^{(2)}$, -0.42, push cart to the left ). |
| Task16 | CartPole | ( $s^{(1)}$, 100, push cart to the right ), ( $s^{(3)}$, -100, push cart to the left ). |
| Task17 | CartPole | ( $s^{(0)}$, -4.8, push cart to the right ), ( $s^{(3)}$, -100, push cart to the left ). |
| Task18 | CartPole | ( $s^{(1)}$, 100, push cart to the right ), ( $s^{(2)}$, -0.42, push cart to the left ). |
| Task19 | CartPole | ( $s^{(0)}$, -4.8, push cart to the right ), ( $s^{(1)}$, 100, push cart to the right ), ( $s^{(2)}$, -0.42, push cart to the left ), ( $s^{(3)}$, -100, push cart to the left ). |
| Task20 | Acrobot | ( $s^{(3)}$, -1, apply -1 torque ), ( $s^{(4)}$, 12.57, apply 0 torque ), ( $s^{(5)}$, 28.27, apply 1 torque ). |
| Task21 | MountainCar | ( $s^{(0)}$, -0.07, not accelerate ), ( $s^{(1)}$, 0.07, accelerate to the right ). |
| Task22 | Pendulum | ( $s^{(2)}$, 8, maximum left torque ), ( $s^{(1)}$, -1, maximum right torque ). |
| Task23 | Pendulum | ( $s^{(2)}$, 8, maximum left torque ), ( $s^{(2)}$, -8, maximum right torque ). |
| Task24 | Pendulum | ( $s^{(1)}$, -1, maximum right torque ), ( $s^{(2)}$, -8, maximum right torque ). |
| Task25 | Pendulum | ( $s^{(2)}$, 8, maximum left torque ), ( $s^{(1)}$, -1, maximum right torque ), ( $s^{(2)}$, -8, maximum right torque ). |
| Task26 | Lunar Lander | ( $s^{(0)}$, 1.5, do nothing ). |
| Task27 | Lunar Lander | ( $s^{(2)}$, -5, fire left orientation engine ). |
| Task28 | Lunar Lander | ( $s^{(4)}$, 3.14, fire main engine ). |
| Task29 | Lunar Lander | ( $s^{(6)}$, 0, fire right orientation engine ). |
| Task30 | Bipedal Walker | ( $s^{(0)}$, 3.14, maximum forward speed ). |
| Task31 | Bipedal Walker | ( $s^{(1)}$, 5, maximum backward speed ). |
| Task32 | Lunar Lander | ( $s^{(0)}$, 1.5, do nothing ), ( $s^{(4)}$, 3.14, fire main engine ). |
| Task33 | Lunar Lander | ( $s^{(2)}$, -5, fire left orientation engine ), ( $s^{(6)}$, 0, fire right orientation engine ). |
| Task34 | Lunar Lander | ( $s^{(0)}$, 1.5, do nothing ), ( $s^{(6)}$, 0, fire right orientation engine ). |
| Task35 | Lunar Lander | ( $s^{(2)}$, -5, fire left orientation engine ), ( $s^{(4)}$, 3.14, fire main engine ). |
| Task36 | Lunar Lander | ( $s^{(0)}$, 1.5, do nothing ), ( $s^{(2)}$, -5, fire left orientation engine ), ( $s^{(4)}$, 3.14, fire main engine ), ( $s^{(6)}$, 0, fire right orientation engine ). |
| Task37 | Bipedal Walker | ( $s^{(0)}$, 3.14, maximum forward speed ), ( $s^{(1)}$, 5, maximum backward speed ). |
| Task38 | Half Cheetah | ( $s^{(1)}$, 5, apply a torque of 1 to all rotors ). |
| Task39 | Half Cheetah | ( $s^{(2)}$, 5, apply a torque of -1 to all rotors ). |
| Task40 | Hopper | ( $s^{(1)}$, 5, apply a torque of 1 to all rotors ). |
| Task41 | Hopper | ( $s^{(2)}$, -5, apply a torque of -1 to all rotors ). |
| Task42 | Reacher | ( $s^{(0)}$, 5, apply a torque of 1 to all rotors ). |
| Task43 | Reacher | ( $s^{(1)}$, -5, apply a torque of -1 to all rotors ). |
| Task44 | Half Cheetah | ( $s^{(1)}$, 5, apply a torque of 1 to all rotors ), ( $s^{(2)}$, 5, apply a torque of -1 to all rotors ). |
| Task45 | Hopper | ( $s^{(1)}$, 5, apply a torque of 1 to all rotors ), ( $s^{(2)}$, -5, apply a torque of -1 to all rotors ). |
| Task46 | Reacher | ( $s^{(0)}$, 5, apply a torque of 1 to all rotors ), ( $s^{(1)}$, -5, apply a torque of -1 to all rotors ). |
| Task47 | Predator-Prey | ( $s^{(4)}$, 0, move left at max speed ). |
| Task48 | Predator-Prey | ( $s^{(5)}$, 0, remain in place ). |
| Task49 | WorldComm | ( $s^{(4)}$, 0, move left at max speed ). |
| Task50 | WorldComm | ( $s^{(5)}$, 0, remain in place ). |

