# OpenReview forum: "Angel or Demon: Investigating the Plasticity Interventions' Impact on Backdoor Threats in Deep Reinforcement Learning"
_ICML.cc/2026/Conference — ICML 2026 regular_

### Official Review · Reviewer_U46u · 2026-03-11

**Soundness:** 3
**Presentation:** 2
**Significance:** 3
**Originality:** 2
**Overall Recommendation:** 4
**Confidence:** 3

**Summary:**

This paper studies how plasticity interventions in deep reinforcement learning affect the effectiveness of backdoor attacks. The authors conduct a large empirical study across multiple tasks, attacks, interventions, and threat models. The results show that most interventions help mitigate backdoor attacks, while SAM tends to amplify them. The paper also analyzes several model properties and proposes a SCC framework to help explain the observed behavior.

**Compliance With Llm Reviewing Policy:**

Affirmed.

**Final Justification:**

My concerns have been addressed.

**Key Questions For Authors:**

1.	Can the authors use SCC as an actual attack procedure and compare it against SAM-only and a strong attack baseline?
2.	Can the authors quantify the proposed sharpness-based detection as a formal detection task? For example, it would be helpful to report ROC/AUC curves, threshold sensitivity, and false-positive rates on benign runs both with and without interventions.
3.	The paper mentions that some attacks were standardized (adding action tampering and removing trigger optimization for BadRL). Could the authors clarify whether the main qualitative conclusions still hold when using the original, unmodified implementations of these attacks?
4.	How robust is the observed SAM effect on its hyperparameters and optimizer configuration?

**Limitations:**

Yes

**Strengths And Weaknesses:**

Strengths:
The paper studies an important and underexplored security problem in reinforcement learning.
The empirical evaluation covers many combinations of tasks, attacks, and interventions.
The observation that SAM may amplify backdoor attacks is interesting.

Weaknesses
Some findings (e.g., the mitigating effects of weight clipping or layer normalization) appear consistent with observations already reported in the backdoor literature, yet the paper does not clearly position these results relative to prior work.

The proposed mechanisms (activation pathway disruption, representation space compression, and backdoor gradient amplification) are supported primarily by correlational evidence such as pathology rankings and illustrative examples. While these analyses suggest possible explanations, they do not establish that the identified pathologies causally drive the observed ASR/BTP changes.

The paper briefly suggests sharpness-based detection as a potential defense direction. Figure 5(c) shows distributional separation in sharpness behavior, but there is no detector, no threshold, no false-positive study, and no comparison to existing defense approaches.

The benchmarking part is the strongest. The SCC and sharpness-detection parts are weaker. In Figure 10, SCC is presented as a framework for robust backdoor injections, but the paper didn’t implement SCC as an attack and never compares it against strong baselines.

---

> ### Author Rebuttal · Authors · 2026-03-31
>
> Hello Reviewer U46u,
>
> We greatly appreciate your recognition of our empirical study and findings.
> We sincerely thank you for your insightful suggestions.
> Below, we address your questions one by one for clarity.
> We use the **symbol #** (e.g., #Table x) to denote references to items in the Supplementary Materials, available via the anonymous link: https://anonymous.4open.science/r/SM4R-3060.
>
> We hope that our responses address your concerns and help reinforce your confidence in this study. We welcome any further comments and would greatly appreciate it if you could consider increasing your overall recommendation and confidence.
>
> ---
>
> **W1**: Limited positioning of findings relative to prior work.
>
> **R1**: Thanks for your suggestion.
>
> We agree that clearly positioning our findings relative to prior work is important.
> In fact, we already highlight connections to prior work in the manuscript.
> For example, **page 6** states: "This resembles certain mitigation strategies in the deep learning backdoor domain [1]," and our discussions on representation space compression and backdoor gradient amplification are informed by [2,3].
>
> Meanwhile, we would like to clarify that this work is the first to systematically identify three intrinsic mechanisms underlying intervention effects on DRL backdoors, distinguishing it from prior work. Rather than analyzing interventions in isolation, we distill their effects into these mechanisms through empirical, analytical, and theoretical insights, providing guidance for future research.
>
> ---
>
> **W2**: Lack of causal evidence.
>
> **R2**: Thanks for your insightful comment.
>
> We agree that our current analysis does not constitute fully controlled causal identification, and we present the proposed mechanisms as mechanistically grounded explanations supported by consistent evidence rather than strict causal claims.
>
> Our analysis builds on diagnostic metrics widely used in prior DRL plasticity literature [4,5], following standard root-cause analysis practices. For *SAM*, we further provide theoretical support (**Appendix G**), linking sharpness-aware updates to backdoor gradient amplification.
>
> To further strengthen the evidence, we conduct ablation studies to better isolate each mechanism.
> Preliminary results (**#Tables 1–3**) are consistent with our findings, providing additional support.
>
> ---
>
> **W3 & W4 & Q1 & Q2**: The benchmarking part is the strongest. The SCC and sharpness-detection parts are weaker.
>
> **R3**: Thanks for your comment.
>
> We clarify that *SCC* and *Sharpness-Based Detection* are positioned as insight-level contributions rather than fully developed methods, distilled from our empirical findings to provide mechanism-driven guidance for future research.
>
> - For *SCC*, it serves as a conceptual framework unifying the three intrinsic mechanisms, with empirical support from **Tables 1 and 9** and **#Table 6**, which show consistent improvements across multiple baselines.
>
> - For *Sharpness-Based Detection*, **Section 5** discusses its limitations and open challenges (e.g., task dependence and confounders), highlighting it as a promising yet open direction rather than a finalized solution.
>
> We will revise the manuscript to clarify this distinction.
>
> ---
>
> **Q3**: Validation under original attack implementations.
>
> **R4**: Thanks for your comment.
>
> We remove the trigger optimization component in BadRL to enable unified reporting across all attack methods in a single table.
> To assess its impact, we additionally include evaluations with and without trigger optimization.
>
> **#Table 4** shows that the trends across the three interventions remain consistent with and without trigger optimization, which only slightly attenuates but does not fundamentally alter their relative effectiveness.
>
> We will further explore additional interventions and expand the discussion in the revised manuscript.
>
> ---
>
> **Q4**: Robustness of the SAM effect.
>
> **R5**: Thanks for your insightful comment.
>
> We conduct ablation studies to investigate the impact of the neighborhood radius on the effect of *SAM*.
> As shown in **#Table 3**, compared to *None* (ASR = 0.184, BTP = 1.000), *SAM* consistently achieves significantly higher ASR (0.291–0.403) while having minimal impact on BTP (0.992–1.000), further corroborating its amplifying effect on backdoor behaviors.
>
> We further examine the hyperparameter sensitivity of *Weight Clipping* and *Spectral Normalization*, with results presented in **#Table 1 and 2**.
>
> We will include additional discussion on this aspect in the revised manuscript.
>
> ---
>
> **References**:
>
> [1] Backdoor learning: A survey. IEEE transactions on neural networks and learning systems, 2024.
>
> [2] Exploring the Orthogonality and Linearity of Backdoor Attacks. IEEE S&P, 2024.
>
> [3] Clibe: detecting dynamic backdoors in transformer-based nlp models. NDSS, 2025.
>
> [4] Understanding Plasticity in Neural Networks. ICML, 2023.
>
> [5] Loss of plasticity in deep continual learning. Nature, 2024.

---

> > ### Author Rebuttal · Reviewer_U46u · 2026-04-03
> >
> > My concerns have been addressed.

---

> > > ### Author Response · Authors · 2026-04-04
> > >
> > > Hello Reviewer U46u,
> > >
> > >
> > > We sincerely thank you for taking the time to review our manuscript and for your insightful comments. We will incorporate all the suggested improvements in the revised version.
> > >
> > > We would greatly appreciate any additional comments that could help improve your confidence in our work.
> > >
> > > Thank you again, and we wish you all the best.
> > >
> > >
> > > Sincerely,
> > >
> > > Authors of Submission #11340

---

### Official Review · Reviewer_vjFP · 2026-03-13

**Soundness:** 3
**Presentation:** 4
**Significance:** 3
**Originality:** 3
**Overall Recommendation:** 5
**Confidence:** 3

**Summary:**

The paper studies how common training techniques in modern reinforcement learning, known as plasticity interventions, affect the success of backdoor attacks. Through a very large set of experiments, the authors find that most of these techniques make backdoors weaker, but one method called SAM actually makes them stronger by reinforcing the hidden trigger behavior. They use these findings to explain why this happens and introduce a framework that describes how backdoors form under different intervention settings.

**Compliance With Llm Reviewing Policy:**

Affirmed.

**Final Justification:**

My concern with respect to impact on only being able to attack models before deployment stands. However, my concerns with respect to the limit of only evaluating SAM under PPO has been addressed by the highlight of the supplementary materials. I will raise my score.

**Key Questions For Authors:**

The paper relies on two existing threat models (training‑time poisoning and post‑training poisoning), both of which ultimately require the victim to adopt attacker‑modified weights. Can the authors clarify the practical scenarios where TM‑Post is realistic, given that it essentially becomes a supply‑chain compromise rather than a true deployment‑time attack?

The experimental results show strong effects of interventions like SAM under PPO. How confident are the authors that these findings generalize to other RL algorithms or architectures beyond the small supplementary tests, and can they comment on the expected stability of these mechanisms across algorithms?

**Limitations:**

yes

**Strengths And Weaknesses:**

Strengths

The paper fills a real gap by examining how widely used plasticity interventions affect backdoor vulnerabilities in DRL, something prior work explicitly did not study.

It provides large‑scale experimental evidence across 14,664 cases, showing clear, consistent patterns such as SAM amplifying backdoors and most other interventions reducing them.

The mechanistic analyses (activation pathway disruption, representation compression, gradient amplification) meaningfully explain why interventions change backdoor behavior rather than just reporting empirical results.


Weaknesses

The threat models used are entirely inherited from prior DRL backdoor literature and do not offer novelty or refinement.
The TM‑Post setting depends on the attacker delivering modified weights to the victim, which limits its practicality and essentially reduces it to a supply‑chain poisoning scenario.

Although the experimental sweep is large, the study focuses mainly on PPO and a fixed set of interventions, leaving uncertainty about how broadly the findings generalize to other RL algorithms or architectures.

---

> ### Author Rebuttal · Authors · 2026-03-31
>
> Hello Reviewer vjFP,
>
> We greatly appreciate your recognition of our motivation and the mechanistic analyses we adopt.
> In addition, we sincerely thanks for your insightful suggestions.
> In the following, we reply to the questions one by one for the convenience of checking.
> We use the **symbol #** (e.g., #Table x, #Figure x) to denote references to items in the Supplementary Materials, available via the anonymous link: https://anonymous.4open.science/r/SM4R-3060.
>
> We hope that our responses address your concerns and help reinforce your confidence in this study. We welcome any further comments and would greatly appreciate it if you could consider increasing your overall recommendation and confidence.
>
> ---
>
> **W1 & Q1**: The threat models used are entirely inherited from prior DRL backdoor literature and do not offer novelty or refinement. The TM-Post setting depends on the attacker delivering modified weights to the victim, which limits its practicality and essentially reduces it to a supply-chain poisoning scenario.
>
> **R1**: Thanks for your insightful suggestion.
>
> We agree that the two threat models are adopted from prior DRL backdoor literature, and existing works largely remain within these settings [1-4].
> Our goal is to systematically study how plasticity interventions interact with existing attack settings, rather than introducing a new threat model. To the best of our knowledge, this is the first work that jointly evaluates both *TM-Scratch* and *TM-Post* at scale, enabling a more complete understanding of intervention effects.
>
> We respectfully clarify that *TM-Post* represents a broader class of realistic scenarios where a trained agent may be modified prior to deployment.
> Examples include third-party model hosting or sharing, downstream fine-tuning or repackaging, and adversarial modification by the original provider.
> Supply-chain attacks can be viewed as a special case of this setting.
>
> Moreover, injecting a backdoor into a well-trained agent is often more convenient and cost-effective than training from scratch, motivating our focus on post-training and multi-backdoor scenarios.
>
> We will revise the manuscript to clarify the scope and practical relevance of both threat models.
>
> ---
>
> **W2 & Q2**: The experimental results show strong effects of interventions like SAM under PPO. How confident are the authors that these findings generalize to other RL algorithms or architectures beyond the small supplementary tests, and can they comment on the expected stability of these mechanisms across algorithms?
>
> **R2**: Thanks for your insightful suggestion. We believe that the current findings exhibit generalizability within a certain scope.
>
> - We apologize for the lack of clarity.
> In fact, **Appendix H** already includes supplementary experiments on **DDPG** and **MADDPG**. Here, PPO represents on-policy, stochastic policy methods, while DDPG and MADDPG cover off-policy, deterministic policy settings.
> Moreover, we evaluate in two multi-agent environments, where DDPG and MADDPG correspond to the *DTDE* and *CTDE* paradigms, respectively.
> The results in **Table 5** are consistent with our main findings, further supporting the generalizability of our findings across different DRL algorithmic architectures.
>
> - Beyond empirical validation, our confidence in generalization comes from the fact that our analysis is mechanism-driven rather than algorithm-specific. In particular, the key effect (e.g., *SAM* amplifying backdoor threats) relies on general properties, such as:
> (i) backdoor signals induce sharp and high-magnitude gradients,
> (ii) SAM explicitly amplifies sharp directions, and
> (iii) DRL training is inherently non-stationary, leading to competition between benign and backdoor pathways.
> These properties are shared by most gradient-based RL methods, suggesting that the observed mechanisms are expected to be broadly stable across algorithms.
>
> - In the rebuttal, we conduct additional ablation studies to examine the impact of intervention hyperparameters.
> Preliminary results in **#Table 1–3** are consistent with the findings in the manuscript.
> For example, compared to *None* (ASR = 0.184, BTP = 1.000), *SAM* consistently achieves significantly higher ASR (0.291–0.403) while having minimal impact on BTP (0.992–1.000), further corroborating its amplifying effect on backdoor behaviors.
> We will further extend and refine these evaluations in future revisions.
>
> - We acknowledge that the generalization is not universal. For example, the mechanism may not directly apply to non-gradient-based methods (e.g., tabular Q-learning).
>
> ---
>
> **References**:
>
> [1] Sleepernets: Universal backdoor poisoning attacks against reinforcement learning agents. NeurIPS 2024.
>
> [2] BadRL: Sparse Targeted Backdoor Attack Against Reinforcement Learning. AAAI, 2024.
>
> [3] Adversarial Inception Backdoor Attacks against Reinforcement Learning. ICML 2025.
>
> [4] TrojanTO: Action-Level Backdoor Attacks against Trajectory Optimization Models. ICLR 2026.

---

> > ### Author Rebuttal · Reviewer_vjFP · 2026-04-05
> >
> > My concern with respect to impact on only being able to attack models before deployment stands. However, my concerns with respect to the limit of only evaluating SAM under PPO has been addressed by the highlight of the supplementary materials. I will raise my score.

---

> > > ### Author Response · Authors · 2026-04-07
> > >
> > > Hello Reviewer vjFP,
> > >
> > > We sincerely thank you for taking the time to review our manuscript and for your insightful comments. We will incorporate all of the suggested improvements in the revised version.
> > >
> > > In addition, to the best of our current understanding, we would like to discuss the deployment-time attack scenario you raised from two perspectives, which we hope will help address your concern:
> > >
> > > (1) The threat models considered in our paper (including *TM-Scratch* and *TM-Post*) can be extended to deployment-time attacks, provided that the following two conditions hold:
> > >
> > > - The victim agent continues to update its policy during deployment based on interactions with the environment.
> > > This setting is widely recognized, as prior work on DRL plasticity explicitly studies the agent's ability to continuously adapt and learn from environmental interactions during deployment [1–5].
> > >
> > > - The adversary can manipulate the transition data during this stage.
> > > More specifically, this requirement can be satisfied in either of two ways: the adversary directly modifies the transitions stored in the victim agent's replay buffer, or the adversary manipulates the transitions by perturbing the environment states and reward signals.
> > > In **Appendix B.2**, we have categorized these as two paradigms of existing DRL backdoor attacks and have discussed them in detail.
> > >
> > > (2) The DRL backdoor attacks discussed in our paper already inherently involve a deployment-time attack component.
> > >
> > > This is because, in order to achieve the attack objective, the adversary must inject the trigger into the environment at specific moments during deployment, thereby manipulating the victim agent's action output.
> > >
> > > To provide a more intuitive illustration, please refer to **#Table 5** in the anonymous link (https://anonymous.4open.science/r/SM4R-3060), which reports task-specific scores and shows how an adversary can cause a DRL agent to fail by manipulating its actions.
> > > For example, in Lunar Lander, the adversary can force the agent to crash rapidly, reducing the average score from 244.13 to −882.97, which is substantially worse than even a random policy (−175.10).
> > >
> > > We plan to add a discussion of this issue in Appendix B.
> > >
> > > We hope that the above clarification from these two perspectives addresses your concern and further strengthens your confidence in the manuscript.
> > >
> > > Thank you again, and we wish you all the best.
> > >
> > > Sincerely,
> > > Authors of Submission #11340
> > >
> > > ---
> > >
> > > **References**:
> > >
> > > [1] Understanding and Preventing Capacity Loss in Reinforcement Learning. ICLR, 2022.
> > >
> > > [2] Understanding Plasticity in Neural Networks. ICML, 2023.
> > >
> > > [3] Revisiting Plasticity in Visual Reinforcement Learning. ICLR, 2024.
> > >
> > > [4] Loss of Plasticity in Deep Continual Learning. Nature, 2024.
> > >
> > > [5] Normalization and Effective Learning Rates in Reinforcement Learning. NeurIPS, 2024.

---

### Official Review · Reviewer_vFYF · 2026-03-14

**Soundness:** 2
**Presentation:** 3
**Significance:** 2
**Originality:** 3
**Overall Recommendation:** 3
**Confidence:** 4

**Summary:**

This paper investigates how plasticity interventions — techniques designed to maintain learning capability in DRL agents — affect the vulnerability of those agents to backdoor attacks. Through a large-scale empirical study of 14,664 cases, the authors find that most interventions mitigate backdoor threats while SAM uniquely amplifies them. Building on these findings, the authors propose SCC, a conceptual framework for robust backdoor injection.

**Compliance With Llm Reviewing Policy:**

Affirmed.

**Final Justification:**

While I acknowledge the impressive experimental scope and the genuinely interesting observations in this paper, I maintain my weak reject. My core concern is that SCC is proposed as a design framework with multiple components, yet the paper's own results demonstrate that combining these components produces non-additive and difficult-to-predict effects — a point the authors themselves acknowledge but do not resolve. Presenting such a framework without empirical validation of the full combination feels premature to me. I note that I seem to hold a different opinion than the other reviewers, and I am happy to defer to the AC on this. In any case, this was a very interesting read.

**Key Questions For Authors:**

Q1: I am curious about the selection criteria for the studied interventions. Standard techniques like gradient noise and clipping are arguably more common in DRL stability and plasticity management than some methods investigated here. Why were these foundational methods excluded from the study?

**Limitations:**

Yes

**Strengths And Weaknesses:**

**Strength**:
1. The problem itself is well-motivated — plasticity interventions are now standard components of DRL systems, yet their security implications have been largely overlooked. This is a genuine blind spot in the literature worth investigating.

2. Extensive attack scenarios have been studied, including four backdoor attacks and 47 backdoor tasks covering both single- and multi-backdoor settings. Multiple interventions—eight in total—have also been evaluated.

3. Some of findings are intriguing. I think the mechanistic insights where the impact of interventions on pathologies are investigated offer some useful insights.  Specially, the observation that SAM amplifies backdoor is surprising

**Weaknesses**:
1. The SCC framework is presented as a robust backdoor attack, but this claim is never empirically validated — there are no experiments showing it achieves better ASR than existing attacks.

2. While the paper includes extensive experiments, I think some conclusions in the "Intrinsic Mechanism" section are drawn a bit too broadly from the observed results.

3. Despite the interesting results, the paper's contribution remains primarily observational. The findings are not translated into actionable outcomes, such as a concrete new attack, a benchmarked defense, or a validated framework.

Overall, I think if the authors had used these observations to introduce and empirically validate a concrete attack or defense, this would have been a much stronger paper.

**Minor Comments**:
1. I think adding the names of the interventions directly in the captions of Figures 3 and 4 would improve readability. I found myself constantly going back and forth between the text and the figures to remember, for example, which intervention corresponds to p3. I would recommend merging Figures 3 and 4 into a single figure with one caption.

---

> ### Author Rebuttal · Authors · 2026-03-31
>
> Hello Reviewer vFYF,
>
> We greatly appreciate your recognition of the scale of our experiments and the clarity of our findings.
> We sincerely thank you for your insightful suggestions.
> Below, we address your questions one by one for clarity.
> We use the **symbol #** (e.g., #Table x, #Figure x) to denote references to items in the Supplementary Materials, available via the anonymous link: https://anonymous.4open.science/r/SM4R-3060.
>
> We hope that our responses address your concerns and help reinforce your confidence in this study. We welcome any further comments and would greatly appreciate it if you could consider increasing your overall recommendation and confidence.
>
> ---
>
> **W1 & W3**: SCC is presented as a robust backdoor attack, but this claim is never empirically validated - there are no experiments showing it achieves better ASR than existing attacks. Despite the interesting results, the paper's contribution remains primarily observational. The findings are not translated into actionable outcomes.
>
> **R1**: Thanks for your comment.
>
> We clarify that *SCC* is not intended as a standalone attack algorithm, but as a conceptual framework distilled from our empirical findings (see **Figure 10**). Its goal is to provide a mechanism-driven design blueprint for constructing robust backdoor attacks, rather than a fully engineered method with task-specific tuning.
>
> While *SCC* itself is conceptual, its effectiveness is indirectly validated through *SCC*-compliant configurations. As shown in **Tables 1 and 9**, across multiple tasks and four existing attacks, configurations that progressively integrate the three *SCC* components (*Plastic*, *SLac*, and *SSW*) consistently improve both ASR (0.178 → 0.418) and BTP (0.745 → 0.915).
> Importantly, these gains are achieved via direct combinations of interventions without customized design or hyperparameter tuning, suggesting that *SCC* provides a meaningful and generalizable design principle with substantial room for further improvement.
>
> Beyond observation, our findings enable concrete and actionable outcomes (see **Section 5.3**) :
> - Attack design: *SCC* provides a structured way to compose interventions based on their intrinsic mechanisms.
> - Intervention selection: Pathological diagnosis offers a principled criterion for combining interventions.
> - Pre-deployment analysis: *SCC* supports diagnosing risks in settings where multiple interventions are jointly applied.
>
> We will revise the manuscript to provide a more detailed discussion of these aspects.
>
> ---
>
> **W2**: While the paper includes extensive experiments, I think some conclusions in the "Intrinsic Mechanism" section are drawn a bit too broadly from the observed results.
>
> **R2**: Thanks for this insightful comment.
>
> We agree that care is needed to avoid over-generalizing from empirical observations.
> Our analysis follows a structured pipeline to mitigate this concern:
>
> - We first identify heterogeneous effects of interventions through large-scale empirical results.
>
> - We then analyze these differences using established interpretability tools from the DRL plasticity domain [1–3].
> To further support these analyses, we provide visualization-based evidence (e.g., **Figures 7, 8, 14, 15**).
>
> - Based on the alignment between empirical patterns and interpretable metrics, we derive the three intrinsic mechanisms.
> Importantly, these mechanisms are not assumed a priori, but emerge from consistent observations.
> Moreover, for **Backdoor Gradient Amplification**, we further provide theoretical analysis (see **Appendix G**), strengthening its validity beyond empirical evidence.
>
> We further conduct ablation studies to examine how the findings relate to intervention hyperparameters.
> The preliminary results (see **#Tables 1–3**) are consistent with our current findings, offering additional support.
> We will continue to expand these ablation studies and include corresponding discussion in the revised manuscript.
>
> ---
>
> **MC**: I would recommend merging Figures 3 and 4 into a single figure with one caption.
>
> **R3**: Thanks for your suggestion.
>
> We have merged **Figures 3 and 4** and updated the text information to improve readability, and the result is shown in **#Figure 1**.
>
> ---
>
> **Q1**: I am curious about the selection criteria for the studied interventions.
>
> **R4**: Thanks for your comment.
>
> We select interventions that have been explicitly proposed to address the plasticity issue in DRL.
> These methods are proposed in major venues and are widely adopted as standard baselines, making them both representative and practically relevant.
> While our study already includes seven diverse interventions, we acknowledge that it is not exhaustive.
> We will extending this line of study in future work.
>
> ---
>
> **References**:
>
> [1] Understanding and Preventing Capacity Loss in Reinforcement Learning. ICLR, 2022.
>
> [2] Understanding Plasticity in Neural Networks. ICML, 2023.
>
> [3] Loss of plasticity in deep continual learning. Nature, 2024.

---

> > ### Author Rebuttal · Reviewer_vFYF · 2026-04-02
> >
> > I appreciate the authors' clarification of SCC as a design blueprint. However, I still wonder why the authors did not take it one step further and instantiate it as a concrete method. This seems like a natural next step, especially given that the paper's own results show that combining interventions produces non-additive effects — meaning the behavior of the full combination cannot simply be inferred from individual components alone.

---

> > > ### Author Response · Authors · 2026-04-02
> > >
> > > Hello Reviewer vFYF,
> > >
> > > We sincerely thanks for your insightful feedback, which is highly valuable for improving our work.
> > >
> > > We agree with your suggestion that developing a concrete method based on our findings is a meaningful and important direction. In fact, this is already part of our ongoing research agenda. We are planning to present *SCC* and *Sharpness-Based Detection* as two separate papers, rather than combining them into the current manuscript, as each direction entails substantial additional work (see **Section 5.3**). More importantly, we believe the current submission already contains three well-defined and indispensable contributions, each corresponding to one of our core research questions, forming a coherent and unified narrative:
> > >
> > > - We conduct a comprehensive empirical study covering mainstream DRL backdoor attacks, threat models, and intervention strategies.
> > >
> > > - Building upon empirical results, interpretability analyses, theoretical justifications, and additional ablation studies, we distill five key findings, and further summarize the effects of interventions into three intrinsic mechanisms.
> > >
> > > - We propose *SCC* and *Sharpness-Based Detection* as two novel insights and discuss their potential directions for future exploration.
> > >
> > > We firmly believe that the current manuscript already provides substantial contributions, especially considering its length (27 pages). Under this premise, we consider it more appropriate to develop the follow-up research separately rather than incorporating all aspects into a single paper.
> > >
> > > In summary, we respectfully ask you to reconsider the possibility of increasing your rating based on the current contributions of the manuscript.
> > >
> > > Regardless of your decision, we sincerely appreciate your time and thoughtful review.
> > >
> > > Sincerely,
> > >
> > > Authors of Submission #11340

---

### Official Review · Reviewer_1Cn3 · 2026-03-17

**Soundness:** 3
**Presentation:** 3
**Significance:** 3
**Originality:** 3
**Overall Recommendation:** 4
**Confidence:** 3

**Summary:**

This paper examines plasticity techniques in deep reinforcement learning (DRL) and how they affect agents' vulnerability to backdoor attacks. The paper runs a large set of experiments over 14,000 cases across different tasks and attack types. The authors find that most of these techniques actually help reduce the effectiveness of backdoor attacks. However, SAM exacerbates it. It consistently makes agents more vulnerable, even in post-training settings. The authors explain these patterns in terms of three mechanisms, activation pathway disruption, representation space compression, and backdoor gradient amplification. They also propose an SCC framework for robust backdoor injection and a sharpness-based detection.

**Compliance With Llm Reviewing Policy:**

Affirmed.

**Final Justification:**

Overall, the rebuttal addressed my main concerns. I am raising my score accordingly.

**Key Questions For Authors:**

Please refer to the weaknesses

**Limitations:**

Please refer to the weaknesses

**Strengths And Weaknesses:**

Strengths:

The paper poses an underexplored question. The plasticity interventions are standard in deployed DRL systems, but their interaction with backdoor vulnerabilities has been ignored.


The empirical results are impressive. They studied 14,664 cases across diverse environments, attack methods, and threat models in multiple seeds.

They also propose the SCC framework that unifies different interventions for understanding robust backdoor injection.

The paper is well structured, and the problem setup is clear.

Weaknesses:

The paper's key contribution is on the observed effects of the three mechanisms, but the claims are more supported by the correlational analyses. I would be great if the author could present the ablations that directly affect these mechanisms.

All experiments are conducted using a single on-policy, which may raise questions about the generalization of those findings to the off-policy algorithm since the plasticity behavior and training dynamics can be significantly different across the RL framework.

The backdoor attacks are based on predefined poisoning rates in the environment. The backdoor is designed to directly perturb a single state dimension to a fixed value.  A learnable trigger would show better generalizability of the findings.

The ASR measures whether a specific trigger produces the target action, but does not capture how damaging that action is in context. The threat level of a backdoor depends on more than ASR.

---

> ### Author Rebuttal · Authors · 2026-03-31
>
> Hello Reviewer 1Cn3,
>
> We appreciate your recognition of our motivation and its solid empirical foundation.
> We sincerely thank you for your insightful suggestions.
> Below, we address your questions one by one for clarity.
> We use the **symbol #** (e.g., #Table x, #Figure x) to denote references to items in the Supplementary Materials, available via the anonymous link: https://anonymous.4open.science/r/SM4R-3060.
>
> We hope that our responses address your concerns and help reinforce your confidence in this study. We welcome any further comments and would greatly appreciate it if you could consider increasing your overall recommendation and confidence.
>
> ---
>
> **W1**: I would be great if the author could present the ablations that directly affect these mechanisms.
>
> **R1**: Thanks for your valuable suggestion.
>
> For the three mechanisms, we select *Weight Clipping*, *Spectral Normalization*, and *SAM* as representative interventions and conduct ablation studies to investigate the effects of the uniform bound (0.1-0.5), power iteration (1-5), and neighborhood radius (0.01-0.05), respectively.
>
> Preliminary results in **#Table 1–3** are consistent with the findings in the manuscript.
> For example, compared to *None* (ASR = 0.184, BTP = 1.000), *SAM* consistently achieves significantly higher ASR (0.291–0.403) while having minimal impact on BTP (0.992–1.000), further corroborating its amplifying effect on backdoor behaviors.
>
> We will further extend these ablation studies and include additional discussion on this aspect in the revised manuscript.
>
> ---
>
> **W2**: All experiments are conducted using a single on-policy, which may raise questions about the generalization of those findings to the off-policy algorithm since the plasticity behavior and training dynamics can be significantly different across the RL framework.
>
> **R2**: Thanks for your insightful comment.
>
> We agree that exploring the generalization of our findings across the RL framework is an important issue.
>
> In fact, **Appendix H** of the original manuscript already includes supplementary experiments on two widely studied off-policy DRL algorithms, **DDPG** and **MADDPG**. Here, PPO represents on-policy, stochastic policy methods, while DDPG and MADDPG cover off-policy, deterministic policy settings.
> Moreover, we evaluate in two multi-agent environments, where DDPG and MADDPG correspond to the *DTDE* and *CTDE* paradigms, respectively.
> The results in **Table 5** are consistent with our main findings, further supporting the generalizability of our findings across different DRL algorithmic architectures.
>
> We will include further discussion on this aspect and clarify the scope of applicability of our findings in the revised manuscript.
>
> ---
>
> **W3**: A learnable trigger would show better generalizability of the findings.
>
> **R3**: Thanks for your valuable suggestion.
>
> We agree with your point and add an investigation of the impact of fixed versus learnable triggers by toggling the trigger optimization component in BadRL [1].
>
> **#Table 4** shows that the trends across the three interventions remain consistent with and without trigger optimization, which only slightly attenuates but does not fundamentally alter their relative effectiveness.
> We attribute this to the fact that trigger optimization alleviates the competitive interference between the backdoor and benign pathways during their formation, thereby mildly reducing the impact of the interventions.
>
> We plan to further explore additional interventions and include a more detailed discussion of this issue in the revised manuscript.
>
> ---
>
> **W4**: The ASR measures whether a specific trigger produces the target action, but does not capture how damaging that action is in context.
>
> **R4**: Thanks for your comment.
>
> Numerous prior works have established that ASR is positively correlated with DRL backdoor risk, which is now widely recognized as a consensus in the DRL backdoor domain [1-5].
> We adopt ASR due to its normalized nature, which facilitates consistent reporting and comparison across different DRL tasks.
>
> To provide an intuitive illustration, we add **#Table 5**, which presents task-specific scores and demonstrates how an adversary can cause a DRL agent to fail by manipulating its actions.
> For example, in Lunar Lander, the adversary forces the agent to crash rapidly, causing the average score to drop from 244.13 to −882.97, which is significantly worse than that of a random policy (−175.10).
>
> ---
>
> **References**:
>
> [1] BadRL: Sparse Targeted Backdoor Attack Against Reinforcement Learning. AAAI, 2024.
>
> [2] Sleepernets: Universal backdoor poisoning attacks against reinforcement learning agents. NeurIPS 2024.
>
> [3] UNIDOOR: A Universal Framework for Action-Level Backdoor Attacks in Deep Reinforcement Learning. arXiv 2025.
>
> [4] Adversarial Inception Backdoor Attacks against Reinforcement Learning. ICML 2025.
>
> [5] TrojanTO: Action-Level Backdoor Attacks against Trajectory Optimization Models. ICLR 2026.

---

> > ### Author Rebuttal · Reviewer_1Cn3 · 2026-04-04
> >
> > Thank you for the rebuttal. The rebuttal addresses all my concerns.

---

> > > ### Author Response · Authors · 2026-04-04
> > >
> > > Hello Reviewer 1Cn3,
> > >
> > > We sincerely thank you for taking the time to review our manuscript and for your valuable and insightful comments. We will carefully incorporate all your suggestions into the revised version.
> > >
> > > We would greatly appreciate any further feedback that could help strengthen the manuscript and increase your confidence in our work.
> > >
> > > Thank you once again for your time and support.
> > >
> > > Sincerely,
> > > The Authors of Submission #11340

---

### Decision · Program_Chairs · 2026-04-30

**Decision:**

Accept (regular)

**Comment:**

The AC has carefully read all the reviews and the authors' responses. The rebuttal addressed many of the concerns. After the rebuttal, Reviewer vFYF still has concerns about SCC and maintains a weak reject, while Reviewer vjFP raised the score to accept but remains concerned about "only being able to attack models before deployment". The other two reviewers acknowledged their main concerns have been addressed and recommend a weak accept. Considering all reviews and the authors' response, this paper clearly has some merits. The recommendation is therefore weak accept. The authors should carefully revise their paper further according to the reviewers' comments.